# Birder: Communication-Efficient 1-bit Adaptive Optimizer for Practical Distributed DNN Training

**Hanyang Peng**[1*], **Shuang Qin**[1*], **Yue Yu**[1†], **Jin Wang**[1], **Hui Wang**[1], **Ge Li**[2†]

[1]Peng Cheng Laboratory, Shenzhen, China
[2]School of Electronic and Computer Engineering,
Shenzhen Graduate School, Peking University, Shenzhen, China
penghy@pcl.ac.cn, qinsh@pcl.ac.cn, yuy@pcl.ac.cn
wangj05@pcl.ac.cn, wangh06@pcl.ac.cn, geli@ece.pku.edu.cn

## Abstract

Various gradient compression algorithms have been proposed to alleviate the communication bottleneck in distributed learning, and they have demonstrated effectiveness in terms of high compression ratios and theoretical low communication complexity. However, when it comes to practically training modern deep neural networks (DNNs), these algorithms have yet to match the inference performance of uncompressed SGD-momentum (SGDM) and adaptive optimizers (*e.g.*, Adam). More importantly, recent studies suggest that these algorithms actually offer no speed advantages over SGDM/Adam when used with common distributed DNN training frameworks ( *e.g.*, *DistributedDataParallel (DDP)*) in the typical settings, due to heavy compression/decompression computation or incompatibility with the efficient *All-Reduce* or the requirement of uncompressed warmup at the early stage. For these reasons, we propose a novel 1-bit adaptive optimizer, dubbed **Bi**nary **r**andomization a**d**aptive optimiz**er** (**Birder**). The quantization of Birder can be easily and lightly computed, and it does not require warmup with its uncompressed version in the beginning. Also, we devise *Hierarchical-1-bit-All-Reduce* to further lower the communication volume. We theoretically prove that it promises the same convergence rate as the Adam. Extensive experiments, conducted on 8 to 64 GPUs (1 to 8 nodes) using *DDP*, demonstrate that Birder achieves comparable inference performance to uncompressed SGDM/Adam, with up to $2.5\times$ speedup for training ResNet-50 and $6.3\times$ speedup for training BERT-Base. Code is publicly available at https://openi.pcl.ac.cn/c2net_optim/Birder.

## 1 Introduction

With the rapid development of computational power, "bigger" and "bigger" deep neural network (DNN) models are proposed for expect better performance, from the early classical models, such as AlexNet(61 million parameters) [15], and ResNet (ResNet-50: 20.5 million parameters) [12] to the current large language models (LLMs), such as BERT (BERT-Lagre: 340 million parameters )[10], and GPT (GPT-3: 176 billion parameters)[5]. Scalable parallelism across distributed computing workers for training these large-scale models becomes a necessity. During training, millions to billions of parameters need to be communicated among workers at each iteration, so distributed large-scale DNN training almost invariably suffers from the communication bottleneck.

To address the communication bottleneck, a wide variety of lossy gradient compression algorithms have been proposed to lower the communication overhead. The algorithms can be broadly divided

---

[*]Equal Contribution
[†]Corresponding Author

into three categories based on the compression techniques, including low-precision approximation (*e.g.*, SignSGD[4], TernGrad [31], and QSGD [3], 1-bit Adam [28]), low-rank simplification (*e.g.*, ATOMO[30], PowerSGD [29], and GradZip [9]), and sparsification (*e.g.*, Random-$k$ [27], Top-$k$ [2], and MSTop-$k$ [24]). In addition to the specific compression techniques, some other works, such as Error Feedback(EF) [23][26], EF21 [21], DIANA [19] and MARINA [11], focus on changing the compression objects from the gradient to the gradient and delayed error summation, or the gradient differences to mitigate compressing errors and/or accelerate the convergence rate.

Gradient compression algorithms have demonstrated promising results with a high compression ratio and low oracle/communication complexity in theory. However, when practically training DNNs, they are still **inferior to** uncompressed SGDM/Adam in terms of inference performance. This is because, these gradient compression algorithms are SGD-type optimizers, which will be commonly reduced to vanilla SGD without momentum if compression is not employed. The performance for a compressed optimizer is commonly upper bounded by its uncompressed counterpart, while vanilla SGD is typically less effective than SGDM for training DNNs. Particularly , SGD-type optimizers are known to be substantially inferior to adaptive optimizers (*e.g.*, Adam) for training Transformer-based networks [17] [34] [7], which have become predominant in the DNN community. *Supporting empirical evidences for this phenomenon can be found in Section B in the Appendix.* Furthermore, if we apply the techniques of gradient compression algorithms to compress and communicate the gradients, and subsequently utilize the compressed gradients to construct adaptive optimizers in the local nodes, the final performance will be degraded [28]. Therefore, designing native communication-compression adaptive optimizers is an underexplored problem that requires further research.

As for the system-level speed, recent studies ([33],[1]) pointed out, when distributedly training typical DNN models (*e.g.*, ResNet-50 and BERT-Base) with off-the-shelf *DistributedDataParallel (DDP)* at typical bandwidths (*e.g.*, 10Gbps), existing gradient compression algorithms are still **slower** than uncompressed SGDM/Adam. This is because, the compressor for these algorithms are either quantization or sparsification or low-rank simplification, which exhibit one or more weaknesses below. ($i$) Vector-wise quantization and low-rank simplification compressors are commonly computationally heavy, and their time cost, in some cases, is close to and even larger than the savings from the reduces communications, as empirical evidence has shown in [33] ; ($ii$) Sparsification compressors and bias quantization compressor are not naively combatable with the efficient communication primitive*All-Reduce* due to their inherent structures, and they have to utilize *All-Gather* for aggregation in stead, which will significantly slow down the communication speed, as empirically shown in [1] ; ($iii$) Some low-rank simplification compressors [29] and quantization compressors [28] [18] need to harness their uncompressed counterparts for warm-up at the early stage to stabilize the convergence, and the warm-up time is commonly nontrivial which to some extent renders their high compression ratios vacuous. Therefore, from a system-level perspective, the design ethos of a system-efficient communication-compression algorithm is that we should guarantee that the compression/decompression of the algorithm is computationally light and takes less time, and it should also be friendly to efficient collective communication primitives. Additionally, there is no need to resort to an uncompressed optimizer for warm-up.

To this end, we propose a 1-bit adaptive optimizer, called **Bi**nary **r**andomization a**d**aptive optimiz**er** (Birder), which use the following updating rule is $x_{t+1} = x_t - \alpha_t \mathcal{Q}\left(\frac{m_t}{b_t}\right)$ where $m_t = \beta m_{t-1} + (1 - \beta)g_t$ , $b_t = \beta b_{t-1} + (1 - \beta)|g_t|$ and $g_t$ is the gradient, and $\mathcal{Q}(\cdot)$ is a element-wise binary quantization operator. The main difference between Birder and existing gradient-quantization algorithms is that we directly quantize the entire adaptive update $\frac{m_t}{b_t}$ rather than quantize the gradient $g_t$ or the momentum $m_t$. Because $-1 \leq \frac{(m_t)_j}{(b_t)_j} \leq 1$, where $(m_t)_j$, $(b_t)_j$ are the $j^{th}$ element of $m_t$, $b_t$ respectively, each element of $\frac{m_t}{b_t}$ is easy to be randomly quantized to $1$ or $-1$ in probability, making the quantization computationally light. Another advantage of Birder is that it does not require a full-precision optimizer to warm up at the early stage to ensure stable convergence. We also demonstrate Birder's convergence rate can match that of Adam. Moreover, taking into accost the nature of Birder, we devise an efficient hierarchical communication scheme to further speed up communication, which sufficiently leverages the ultra-high intra-bandwidth among GPUs within the same node.

In particular, we make the following key **contributions**:

- We propose a novel 1-bit optimizer, dubbed Birder, **which is a native communication-compression *adaptive* algorithm that *element-wise* quantizes the entire model update**

**and does not need to leverage its uncompressed counterpart for warm-up**, making compression/decompression computationally light and the extreme quantization ratio exert its best function (Section 2).

- We theoretically prove that despite emoling extreme 1-bit quantization is employed, **Birder still promise the same convergence speed as the full-precision Adam** (Section 3).

- We develop a new communication scheme for 1-bit communication, called *Hierarchical-1-bit-All-Reduce*, **which sufficiently harnesses the ultra-fast intra-connects to accelerate the local communication, and utilize more efficient commutation primitives to further reduce the communication overhead** (Section 4).

- We perform extensive distributed training experiments to demonstrate the effectiveness of the proposed algorithm. **As far as we know, running with *DDP*, our algorithm is the first work to consistently trump SGDM/Adam in terms of entire running time at little/no inference performance cost**, reaching up to $2.47\times$ speedup for ResNet-50 and $6.26\times$ speedup for BERT-Base on $64$ GPUs. (Section 5).

## 2 One-Bit Adaptive Optimizer Birder

In this section, we focus on solving the following problem for distributed training :

$$\min_{x\in\mathbb{R}^d} f(x) = \frac{1}{n}\sum_{i=1}^{n} f_i(x;\xi^{(i)}) \tag{1}$$

where $x$ is the $d$-dimensional model parameter, $n$ is the number of distributed workers. $\xi^{(i)}$ is the sampled min-batch data on the $i$-the worker. The sampled min-batch data on all the workers is independent and identically distributed (*i.i.d.*). $f_i(x;\xi^{(i)})$ is the loss function. Note that $f_i(x;\xi_i)$ is commonly abbreviated as $f_i(x)$ in the following.

When distributedly training large-scale DNN models, using vanilla full-precision optimizers can cause communication bottleneck issues in gradient communication among workers at each iteration. To alleviate this problem, elegant SignSGD[4] was proposed, which merely takes the sign of each coordinate of the gradients. While this algorithm can substantially reduce the communication overhead, its practical performance is still inferior to popular adaptive optimizers, such as Adam. Fortunately, we observe that the mathematical formulations of SignSGD and Adam have close connections, providing an opportunity to propose a new optimizer that can combine their merits. This new optimizer can considerably reducing the communication volume with light computation, while maintaining fast convergence speed and high inference performance.

The mathematical updating rule of SignSGD can be formulated as:

$$x_{t+1} \leftarrow x_t - \alpha_t \mathrm{Sign}(g_t) = x_t - \alpha_t \frac{g_t}{|g_t|} \tag{2}$$

where $\alpha_t$ is the learning rate, $g_t$ denotes the estimated unbias noisy gradient of $f(x_t)$ with random samples, $\mathrm{Sign}(\cdot)$ is a element-wise signum, and $|\cdot|$ is an element-wise absolute operator.

Whereas the updating rule of vanilla Adam [14] can be expressed as:

$$
\begin{aligned}
m_t &\leftarrow \beta_1 m_{t-1} + (1-\beta_1)g_t, \\
v_t &\leftarrow \beta_2 v_{t-1} + (1-\beta_2)g_t^2, \\
x_{t+1} &\leftarrow x_t - \alpha_t \frac{m_t}{\sqrt{v_t}},
\end{aligned}
\tag{3}
$$

where $\beta_1$ and $\beta_2$ represents the exponential moving average factors [3].

If taking $\beta_1$ and $\beta_2$ to zero, $\beta_1, \beta_2 \to 0$ in Eq. (3), Adam will be reduced to SignSGD.

---

[3]For simplicity, we omit the bias correction for $m_t$ and $v_t$ and the small constant in the numerator.

Given the observations above, we propose a new optimizer that is an intermediate between SignSGD and Adam, referred to as Birder, *i.e.*,

$$
\begin{aligned}
m_t &\leftarrow \beta m_{t-1} + (1-\beta) g_t, \\
b_t &\leftarrow \beta b_{t-1} + (1-\beta) |g_t|, \\
x_{t+1} &\leftarrow x_t - \alpha_t \mathcal{Q}\left(\frac{m_t}{b_t}\right),
\end{aligned}
\tag{4}
$$

where the $j$-th elements of $m_t, b_t$ rigorously satisfies $-1 \le \frac{(m_t)_j}{(b_t)_j} \le 1$, $\mathcal{Q}(\cdot)$ is an element-wise quantization operator, and it quantizes the $j$-th element of $\frac{m_t}{b_t}$ as follows:

$$
\mathcal{Q}\left(\frac{(m_t)_j}{(b_t)_j}\right) = \begin{cases} 1, & \text{with probability } p = \frac{1}{2}\left(\frac{(m_t)_j}{(b_t)_j} + 1\right) \\ -1, & \text{with probability } 1-p \end{cases},
\tag{5}
$$

where $\mathbb{E}\left(\mathcal{Q}\left(\frac{m_t}{b_t}\right)\right) = \frac{m_t}{b_t}$, so $\mathcal{Q}(\cdot)$ is unbiased, and proof is provided in Section A of the appendix. The detailed implementation of Birder in a parameter-server model is illustrated in Algorithm 1.

---

**Algorithm 1.** Birder

---

1: **Input**: all workers's model parameter $x_0, x_1$, the $i^{th}$ worker's momentum $m_0^{(i)} = 0$, $b_0^{(i)} = 0$, the $i^{th}$ worker's local error $e_0^{(i)} = 0$, server's global error $\bar{e}_0 = 0$, exponential moving average factor $\beta$, the threshold $T_0$, and the learning rate sequence $\{\alpha_t\}$.
2: **for** $t = 1, ..., T$ **do**
3:     (**On the $i^{th}$ worker**)
4:     Randomly sample $\xi_t^{(i)}$ and compute local gradient: $g_t^{(i)} = \nabla f_i(x_t; \xi_t^{(i)})$
5:     Update the local $m_t^{(i)}$: $m_t^{(i)} = \beta m_{t-1}^{(i)} + (1-\beta) g_t^{(i)}$
6:     Update the local $\hat{b}_t^{(i)}$: $\hat{b}_t^{(i)} = \beta \hat{b}_{t-1}^{(i)} + (1-\beta)|g_t^{(i)}|$
7:     Update the local $b_t^{(i)}$: **if** $t > T_0$ { $b_t^{(i)} = \max(b_{t-1}^{(i)}, \hat{b}_t^{(i)})$} **else** {$b_t^{(i)} = \hat{b}_t^{(i)}$} *
8:     Quantize the local update: $u_t^{(i)} = \mathcal{Q}(\frac{m_t^{(i)}}{b_t^{(i)}} + e_{t-1}^{(i)})$
9:     Update the local error feedback $e_t^{(i)}$ : $e_t^{(i)} = e_{t-1}^{(i)} + \frac{m_t^{(i)}}{b_t^{(i)}} - u_t^{(i)}$
10:     Send $u_t^{(i)}$ to the server
11:     (**On server**)
12:     Average all received $q_t$ and quantize it: $\bar{u}_t = \mathcal{Q}(\frac{1}{n}\sum_{i=1}^n u_t^{(i)} + \bar{e}_{t-1})$
13:     Update the global error feedback $\bar{e}_t$ : $\bar{e}_t = \bar{e}_{t-1} + \frac{1}{n}\sum_{i=1}^n u_t^{(i)} - \bar{u}_t$
14:     Send back $\bar{u}_t$ to all workers
15:     (**On the $i^{th}$ worker**)
16:     Update the local model parameter $x_{t+1}$: $x_{t+1} = x_t - \alpha_t \bar{u}_t$
17: **end for**

---

\* This step follows the technique in AMSGrad [20]. It is more about theoretical significance, and we commonly do not implement it in practice.

The appealing characters of Birder are summarized in the following:

- Compared to SGD-type optimizers, Adam provides a fast convergence rate in practice by adaptively preconditioning the gradients with $v_t$. **Birder inherits this feature to accelerate convergence speed.** On the other hand, unlike Adam, Birder employs the same exponential moving average factor $\beta$ for both $m_t$ and $b_t$. **This eliminates the need for bias correction and reduces the amount of tuning work required.**

- Existing quantization optimizers are built upon *vector-wise quantization* [3][19][22][4] Besides the quantization, they also require heavy computations to compute the norm of a vector

---

[4] The typical gradient-quantized optimizer *QSGD* quantizes the gradient as follows:

$$
\mathcal{Q}((g_t)_j) = \begin{cases} \|g_t\|_p \text{sign}((g_t)_j) \cdot \frac{r}{s}, & \text{with probability } p_i = \frac{s|(g_t)_j|}{\|g_t\|_p} - r \\ \|g_t\|_p \text{sign}((g_t)_j) \cdot \frac{r+1}{s}, & \text{with probability } 1-p_i \end{cases}
$$

where $\|g_t\|_p = (\sum_j |(g_t)_j|^p)^{\frac{1}{p}} (p \ge 1), 0 \le r < s \ (r, l \in \mathbb{N})$ and $\frac{s|(g_t)_j|}{\|g_t\|_2} \in [\frac{r}{s}, \frac{r+1}{s}]$.

and estimate the sign of each element, which will renders the saved communication time cost somewhat meaningless. In contrast, Birder *element-wise* quantizes the update, and the subtle design for $m_t$ and $b_t$ ensures the unquantized update is strictly bounded in the range $[-1, 1]$, **allowing quantization is computed easily and lightly**. Further, unlike most quantization optimizers that only compress the gradients [6], **Birder performs the quantization for the entire adaptive update**, which further streamlines the optimization process.

**Remark.** We have noticed that the prior works 1-bit Adam ([28]) and its variants ([16], [18]) are also categorized as 1-bit adaptive optimizers. However, the design ethos of 1-bit Adam and Birder differ significantly. 1-bit Adam is still built on gradient quantization and and essentially functions as a preconditioned SGDM. 1-bit Adam runs full-precision Adam in the beginning (warm-up phase) and utilizes it as a fixed precondition for SGDM during the rest of training (compression phase). There are three aspects that influence 1-bit Adam to indeed accelerate communication. *First*, the warm-up stage constitutes approximately 15%-25% of the total steps, which to some extent discounts the high quantization ratio. *Second*, the vector-wise quantization employed by 1-bit Adam necessitates extra computations, including the calculation of vector norms and estimation of element signs, which are then transmitted as additional data. These factors diminish the time savings achieved through reduced communication bits. *Third*, the vector-wise quantization technique employed by 1-bit Adam is not compatible with the common-used distributed framework *DDP* (system-level engineered distributed framework). In *DDP*, communication data is uniformly divided into buckets of equal size on the sender's side to enhance communication efficiency. Consequently, when vector-wise quantization is used, communication data from a single layer may be divided into different buckets, resulting in substantial errors during restoration on the receiver's end.

## 3 Theoretical Analysis

In this section, we present the theoretical convergence guarantee for Birder (Algorithm 1). We first introduce some necessary assumptions.

**Assumption 1.**[Bounded infimum] *For any $x$ and a constant $f^*$, we have the objective value $f(x) \geq f^*$.*

**Assumption 2.** [Lipschitz continuous gradient] *The gradient $\nabla f(\cdot)$ is L-Lipschitz continuous, i.e., , $\|\nabla f(x) - \nabla f(y)\| \leq L\|x - y\|_2, \quad \forall x, y \in \mathbb{R}^d$.*

**Assumption 3.** [Unbias and indpendent noisy gradient] *The gradient with respect to the random samples on each worker and at a different time is independent identically distributed (i.i.d.), i.e., $\mathbb{E}[g_t^{(i)}] = \nabla f(x_t), \forall t \geq 1$, $g_t^{(i)}$ is independent of $g_t^{(j)}$ for $i \neq j$, and $g_{t_1}^{(i)}$ is independent of $g_{t_2}^{(j)}$ for $t_1 \neq t_2$.*

**Assumption 4.** [Bounded gradient] *The noisy gradient and the full-set gradient are bounded i.e., $\|g_t^{(i)}\| \leq G, \quad \|\nabla f_t(x)\| \leq G, \quad \forall t \geq 1$.*

Under the assumptions above, we then present the theoretical convergence for Birder in Algorithm 1.

**Theorem 1.** *For Birder in Algorithm 1, under Assumption 1-4, assuming $(b_t^{(i)})_j \geq \rho > 0$ , $\forall j \in [1, 2, ..., d]$[5], choosing $\alpha_t = \frac{c}{\sqrt{t}}$, $\forall t \in [1, 2, ..., T]$ and $\alpha_0 = \alpha_1$ , and defining $z_1 = x_1 + \alpha_1(\delta_1 - e_1)$ where $\delta_1 = \frac{1}{n}\sum_{i=1}^n \frac{m_1^{(i)}}{b_1^{(i)}} - \frac{\sum_{i=1}^n m_1^{(i)}}{\sum_{i=1}^n b_1^{(i)}}$ and $e_1 = \frac{1}{n}\sum_{i=1}^n e_1^{(i)} + \bar{e}_1$, we then have the following*

$$\mathbb{E}\left[\frac{1}{T}\sum_{t=1}^T \|\nabla f(x_t)\|\right]^2 \leq \frac{C_1}{\sqrt{T}} + \frac{C_2(1 + \log T)}{\sqrt{T}}, \tag{6}$$

*where*

$$C_1 = cG\left(\mathbb{E}[f(z_1) - f^*] + \frac{3c^2 dL}{16} + \frac{\beta cdG^2}{(1-\beta)\rho} + \frac{4cdG^2}{\rho} + \frac{c^2\beta^2 LG^2 d}{\rho^2(1-\beta)^2}\right),$$

$$C_2 = c^3 G\left(\frac{(8\beta^2 + 10\beta + 5)L^2 d}{(1-\beta)^2} + \frac{G^2(1+L)}{2\rho^2} + 2dL\right).$$

---

[5]We commonly add a small constant to $b_t$ to avoid zero denominators for numerical stability, which guarantees this assumption holds in practice.

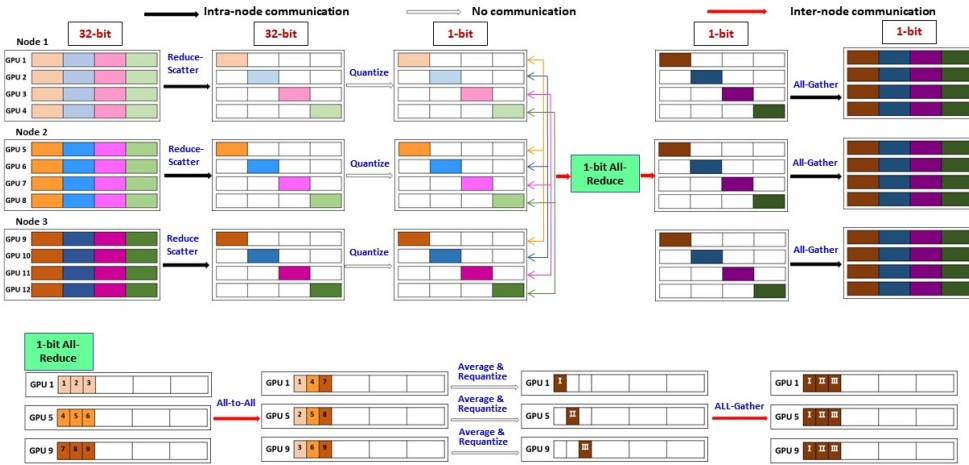

Figure 1: Paradigm of *Hierarchical-1-bit-All-Reduce*

The theoretical results suggested that 1-bit Birder essentially achieve the same convergence rate $(O(\frac{1}{\sqrt{T}}))$ as the uncompressed Adam [14][8].

## 4 Hierarchical-1-bit-All-Reduce

The data communication for Birder is one-bit, which cannot be directly aggregated using the efficient *All-Reduce*. Additionally, there is a significant imbalance between the intra-node and inter-node bandwidths. If we attempt to aggregate the data uniformly from both intra-nodes and inter-nodes, the communication process will be hindered by the inter-node data exchanges, resulting in slower overall communication.

In light of the problems above, we propose a hierarchical communication scheme called *Hierarchical-1-bit-All-Reduce*. This scheme efficiently aggregates our 1-bit data by leveraging the ultra-high intra-node bandwidth and reducing the inter-node communication overhead. Assuming we have $n$ nodes, each containing $m$ GPUs, and the overall volume for each GPU needs to be communicated is $P$,as visually depicted in Figure 1, the steps of *Hierarchical-1-bit-All-Reduce* are as follows: $(i)$ Each GPU performs *Reduce-Scatter* to locally aggregate and scatter the data within its node. The communication volume for each GPU in this step is $\frac{(m-1)P}{m}$. $(ii)$Each GPU then applies Birder to quantize the data, resulting in a reduced volume of $\frac{P}{32m}$ on each GPU. $(iii)$ The GPU proceeds with *1-bit-All-Reduce* to inter-aggregate the data. This step consists of two sub-steps: 1) Each GPU performs *All-to-All* to collect the corresponding data from GPUs in other nodes, with a communication volume of $\frac{(n-1)P}{32mn}$ . 2) Each GPU averages and re-quantizes the data, followed by *All-Gather* operation to gather the data. The communication volume in this sub-step is also $\frac{(n-1)P}{32mn}$. (iv)Finally, each GPU performs *All-Gather* to intra-aggregate the data, with a communication volume of $\frac{(m-1)P}{32m}$.

Compared to the time cost of inter-node communication, the time cost of intra-node communication is relatively insignificant. Thus, when utilizing *Hierarchical-1-bit-All-Reduce*, the majority of the communication cost arises from the *1-bit All-Reduce* step in Step $(iii)$. The communication volume across nodes for all GPUs in this scheme is approximately $\frac{2(n-1)P}{32}$. In contrast, if we were to simply employ the original *All-Gather* to aggregate data, the communication volume across nodes for all GPUs would be approximately $\frac{m^2n(n-1)P}{32}$. Consequently, *Hierarchical-1-bit-All-Reduce* proves significantly more efficient than the original *All-Gather*.

Notably, the work [32] also introduces a 1-bit data communication scheme among nodes, but it can only guarantee the expected value of the gathered 1-bit data equals to the average of the original 1-bit data among nodes, thereby it will bring performance deterioration. In contrast, *All-to-All* in *Hierarchical-1-bit-All-Reduce* ensures the final data exactly equals to the average of the original data.

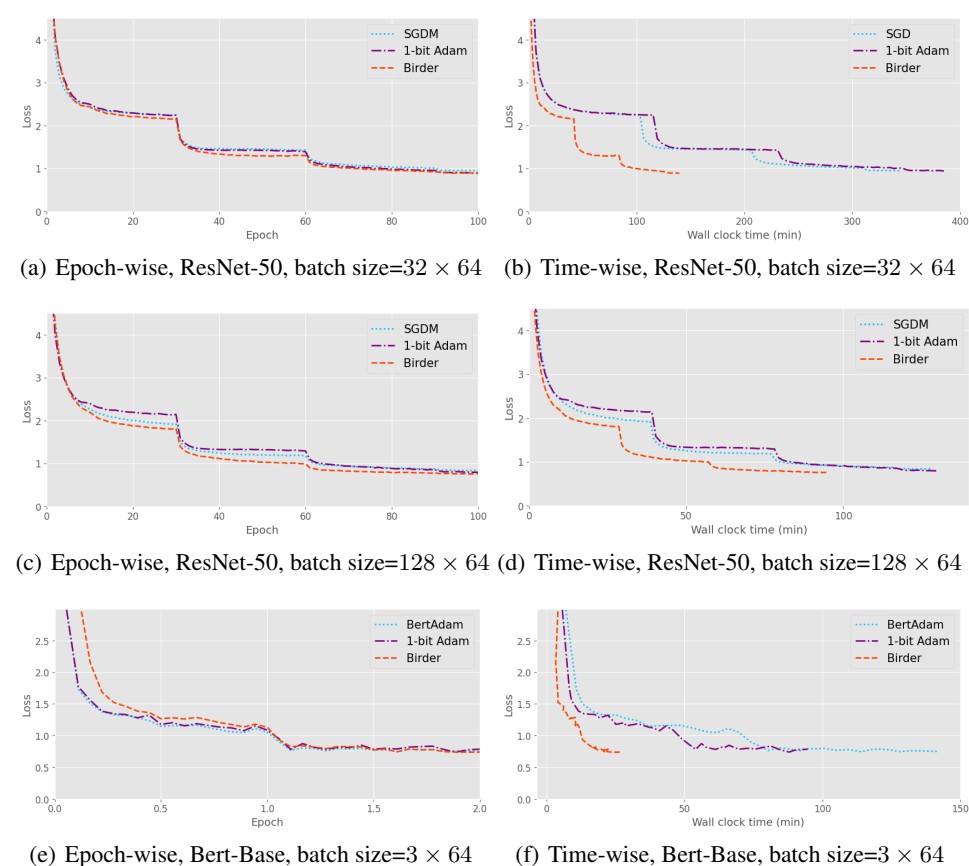

(a) Epoch-wise, ResNet-50, batch size=$32 \times 64$  (b) Time-wise, ResNet-50, batch size=$32 \times 64$

(c) Epoch-wise, ResNet-50, batch size=$128 \times 64$  (d) Time-wise, ResNet-50, batch size=$128 \times 64$

(e) Epoch-wise, Bert-Base, batch size=$3 \times 64$  (f) Time-wise, Bert-Base, batch size=$3 \times 64$

Figure 2: Epoch-wise and time-wise convergence speed for training ResNet-50 with 32 samples per GPU, ResNet-50 with 128 samples per GPU, and fine tuning BERT-Base with 3 samples per GPU with 64 GPUs.

## 5 Experiments

Table 1: System throughput and Test Accuracy of SGDM, 1-bit Adam and Birder for training ResNet-50 on ILSVRC2012 from scratch with $8, 16, 32, 64$ GPUs.

| Optimizer | #GPUs | 32 samples per GPU | | 128 samples per GPU | |
|---|---|---|---|---|---|
| | | Throughput (samples / s) | Top-1 Acc. (%) | Throughput (samples / s) | Top-1 Acc.(%) |
| SGDM | | **3693** (**1.00**×) | 76.19 | **5272** (**1.00**×) | 75.05 |
| 1-bit Adam | 8 | 3243 (0.83×) | 75.55 | 5229 (0.99×) | 75.42 |
| Birder | | 3462 (0.94×) | 75.98 | 5251 (0.99×) | 75.45 |
| SGDM | | 2959 (1.00×) | 75.96 | 6189 (1.00×) | 74.61 |
| 1-bit Adam | 16 | 4745 (1.60×) | 75.33 | 8836 (1.42×) | 75.05 |
| Birder | | **6015** (**2.03**×) | 75.53 | **9633** (**1.56**×) | 75.09 |
| SGDM | | 4270 (1.00×) | 75.47 | 9909 (1.00×) | 74.54 |
| 1-bit Adam | 32 | 7268 (1.70×) | 75.18 | 13827 (1.40×) | 74.62 |
| Birder | | **9416** (**2.21**×) | 75.27 | **15950** (**1.61**×) | 74.82 |
| SGDM | | 6189 (1.00×) | 75.37 | 16640 (1.00×) | 74.22 |
| 1-bit Adam | 64 | 5546 (0.89×) | 75.54 | 16426 (0.99×) | 74.14 |
| Birder | | **15253** (**2.47**×) | 75.30 | **23727** (**1.43**×) | 74.24 |

Recently, several works [33],[1] have shown that when utilizing the system-level engineered distributed data-parallel framework *DDP*, the existing communication-compression optimizers (excluding 1-bit Adam) still perform slower than the uncompressed SGDM/Adam. Therefore, in our evaluation, we focus on assessing the performance of Birder, the uncompressed SGDM/Adam, and the closely related algorithm 1-bit Adam through distributed training experiments using the benchmark models ResNet-50 (CNN) and BERT-Base (Transformer). **More extensive experiments can be founded in Section B of the appendix.**

Table 2: System throughput and F1-Score / Excat-Match of BertAdam, 1-bit Adam and Birder for fine tuning BERT-base on SQuAD 1.1 with $8, 16, 32, 64$ GPUs.

| Optimizer | #GPUs | Throughput (samples/s) | F1-Score (%) | Exact Match (%) |
|---|---|---|---|---|
| BertAdam | | **413** (**1.00**×) | 88.13 | 80.59 |
| 1-bit Adam | 8 | 358 (0.87×) | 88.05 | 80.06 |
| Birder | | 412 (1.00×) | 88.71 | 81.18 |
| BertAdam | | 84 (1.00×) | 88.47 | 81.07 |
| 1-bit Adam | 16 | 213 (2.54×) | 87.87 | 80.31 |
| Birder | | **431** (**5.13**×) | 88.31 | 80.80 |
| BertAdam | | 119 (1.00×) | 88.38 | 80.94 |
| 1-bit Adam | 32 | 274 (2.30×) | 87.78 | 80.08 |
| Birder | | **730** (**6.13**×) | 88.08 | 80.50 |
| BertAdam | | 158 (1.00×) | 88.13 | 80.94 |
| 1-bit Adam | 64 | 252 (1.59×) | 87.33 | 79.67 |
| Birder | | **990** (**6.26**×) | 88.28 | 80.75 |

## 5.1 Experimental Settings

Our experiments were conducted on a testbed consisting of 1, 2, 4, 8 nodes interconnected via 10Gbps Ethernet. Each node was equipped with 8 Nvidia Tesla A100-80GB GPUs. The hardware and software configurations were identical across all instances, with Ubuntu 20.04.4 LTS serving as the operating system. PyTorch 1.11.0 was used as the primary framework, accompanied by CUDA-11.6, cuDNN-8.2, NCCL-2.10.3, and PyTorch 1.11.0 for other relevant libraries. Notably, to ensure compatibility with PyTorch's *DDP*, certain components of Birder and our hierarchical communication scheme were implemented within the customized communication hook of *DDP*.

**Training details.** For the experiments over ResNet-50, we evaluate the convergence and performance of SGDM, 1-bit Adam and Birder on ILSVRC2012. The batch size per GPU is set to 32 or 128 with the standard input resolution $224 \times 224$. When employing *SGDM (baseline)*, the learning rate starts at $0.1 \times \frac{batch\ size}{256}$ with momentum of 0.9 and weight decay of 0.0001. When employing 1-bit Adam and Birder, the learning rate starts at $0.001 \times \frac{batch\ size}{256}$ with weight decay of 0.0001, and $[\beta_1, \beta_2]$ for 1-bit Adam is set to $[0.9, 0.999]$ and $\beta$ for Birder is set to 0.95. Then, the learning rate is divided by 10 after 30, 60 and 90 epochs, and training is finally terminated after 100 epochs. Specifically, the first 15 epochs are used as the warmup stage for 1-bit Adam. For the experiments over BERT-Base, we access the convergence and performance of BertAdam (baseline), 1-bit Adam and Birder for SQuAD 1.1 fine-tuning task using a pre-trained BERT-Base model checkpoint from HuggingFace [6]. The batch size per GPU is set to 3. We perform fine-tuning for 2 epochs. The learning rate linearly increases to $1 \times 10^{-4}$ steps in the early 500 steps and then linearly decreases to 0 in the rest iteration. Specifically, the first $0.2\times$ steps are used as the warmup stage for 1-bit Adam. $[\beta_1, \beta_2]$ for BertAdam, and 1-bit Adam is set to $[0.9, 0.999]$ and $\beta$ for Birder is set to 0.9.

## 5.2 Experimental Results

Figure 2 shows the convergence behaviors of epoch-wise and time-wise training for SGDM / BertAdam (baseline), 1-bit Adam, and Birder using ResNet-50 and BERT-Base models running on 64 GPUs. The experimental results clearly demonstrate that Birder achieves a similar epoch-wise convergence rate compared to the baseline. However, the actual training speed of Birder surpasses both the baseline and 1-bit Adam by a significant margin.

Figure 3 illustrates the system throughput of different optimizers when running ResNet-50 and BERT-Base on 8 GPUs to 64 GPUs (1 node to 8 nodes). When training on 8 GPUs (1 node), where computation takes precedence over communication, the throughput of Birder is slightly lower than that of SGDM and BertAdam. However, as the number of GPUs increases, Birder consistently outperforms its counterparts, and this superiority becomes increasingly evident with more GPUs. Additionally, the system throughput for SGDM, BertAdam, and 1-bit Adam occasionally decreases as the number of GPUs increases, whereas the throughput of Birder steadily grows. This observation indicates that Birder offers better scalability efficiency.

---

[6]https://github.com/huggingface/transformers

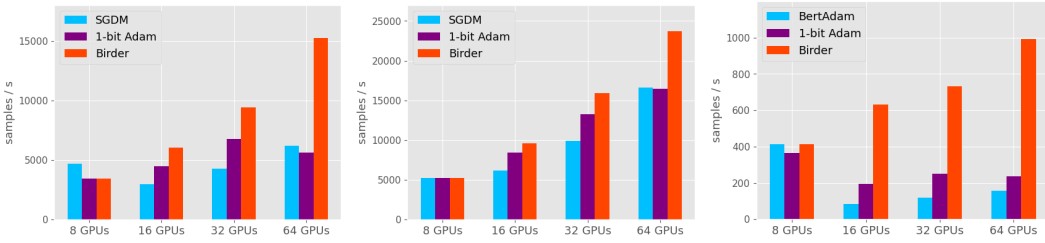

(a) ResNet-50, 32 samples / GPU    (b) ResNet-50, 128 samples / GPU    (c) BERT-Base, 3 samples / GPU

Figure 3: System throughput of optimizers for training (a) ResNet-50 with 32 samples per GPU, (b)ResNet-50 with 128 samples per GPU, and (c) fine tuning BERT-Base with 3 samples per GPU with 8, 16, 32, 64 GPUs.

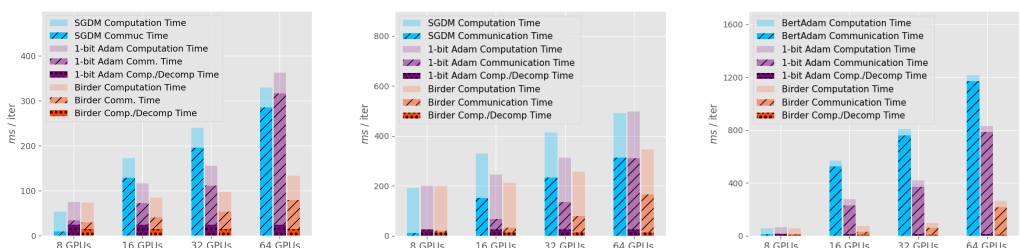

(a) ResNet-50, 32 samples / GPU    (b) ResNet-50, 128 samples / GPU    (c) BERT-Base, 3 samples / GPU

Figure 4: Computation time, communication time and compression/decompression time per iteration of optimizers for training (a) ResNet-50 with 32 samples per GPU, (b)ResNet-50 with 128 samples per GPU, and (c) fine tuning BERT-Base with 3 samples per GPU with 8, 16, 32, 64 GPUs.

In terms of inference performance for ResNet-50, we evaluate the Top-1 accuracy after training on ILSVRC2012 from scratch. For BERT-Base, we measure the F1-score and exact-match score after fine-tuning on SQuAD 1.1. Table 1 shows that when the batch size is set to 32 samples per GPU, the accuracy of Birder is slightly lower than that of SGDM. It has been suggested in some works ([13], [35]) that adaptive optimizers generally yield worse generalization compared to SGDM for CNN architectures. However, as the batch size increases (Table 2), both 1-bit Adam and Birder achieve better accuracy. This can be attributed to the beneficial effect of introducing a certain level of noise for generalization ([25]), which biases the optimizer towards wider valleys. Table 2 demonstrates that Birder achieves similar or higher F1-score and exact-match score compared to BertAdam and 1-bit Adam, validating the effectiveness of Birder for inference tasks.

## 5.3 Communication Efficiency Analysis

As shown in Figure 4, training on a single node demonstrates that the baseline SGDM and BertAdam algorithms are slightly faster compared to Birder and 1-bit Adam. In this scenario, the inter-GPU bandwidth within a node is extremely high, rendering communication time negligible. However, the newly introduced compression/decompression process by Birder and 1-bit Adam adds extra time due to its implementation. Thanks to light-computation quantization, the compression/decompression time for Birder with ResNet-50 and BERT-Base is significantly reduced to approximately $15ms$ and $8ms$ respectively.When conducting distributed training across two nodes, the bandwidth between them is relatively limited (10Gbps in our experiment), making communication time a critical factor. In the case of uncompressed SGDM and BertAdam, the communication time substantially exceeds the computation time for ResNet with 32 samples per GPU and BERT-Base. Consequently, the system throughput is lower compared to a single node (as depicted in Figure 2). However, the extreme 1-bit quantization implemented in Birder effectively reduces communication overhead, resulting in only a marginal increase in the total time required for Birder. As the number of nodes continues to increase, the importance of an efficient communication scheme becomes paramount. By leveraging

our proposed *Hierarchical-1-bit-All-Reduce*, the overall inter-node communication volume exchanged scales proportionally with the number of nodes. In contrast, the *Compressed-All-Reduce* method employed by 1-bit Adam [28] results in the overall communication volume exchanged among nodes being proportional to the number of GPUs (eight times larger than the number of nodes in our experiments). Consequently, as the number of nodes increases, the communication time for Birder exhibits a gradual rise, while the communication time for 1-bit Adam experiences a sudden surge.

## 6  Conclusion

In this study, we introduce a novel 1-bit adaptive optimizer for distributed training. Our optimizer offers the advantages of being lightweight in terms of computation while employing extreme 1-bit quantization for the communication data. Furthermore, we provide theoretical evidence demonstrating that Birder can achieve convergence rates comparable to the uncompressed Adam. To enhance communication speed, we propose a novel communication scheme tailored specifically for Birder, replacing the inefficient naive *All-Gather* approach. Through extensive experiments on benchmark models such as ResNet-50 and BERT-Base, we validate the effectiveness and efficiency of Birder in comparison to uncompressed methods like SGDM/Adam as well as the relevant 1-bit Adam.

## Acknowledgments

This work is supported by the National R&D Program of China (Grant No. 2022ZD0115301), the Major Key Project of PCL (Grant No. PCL2023AS7-1), and the National Natural Science Foundation of China (Grant No. 61806128).

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

# Appendix

## A   Theoretical Analysis for Algorithm 1

In practice, we implement Birder in a non-parameter-server model to further reduce the communication overhead, but the data exchange is essentially equivalent to that in a parameter-server prototype. Hence, we provide the theoretical analysis for Birder in a parameter-server model as shown in Algorithm 1.

According to Algorithm 1, the update $\bar{u}_t$ can be recursively formulated as

$$
\begin{aligned}
\bar{u}_t =& \mathcal{Q}\left(\frac{1}{n}\sum_{i=1}^{n} u_t^{(i)} + \bar{e}_t\right) \\
=& \frac{1}{n}\sum_{i=1}^{n} u_t^{(i)} + \bar{e}_t - \bar{e}_{t+1} \\
=& \frac{1}{n}\sum_{i=1}^{n} \mathcal{Q}\left(\frac{m_t^{(i)}}{b_t^{(i)}} + e_t^{(i)}\right) + \bar{e}_t - \bar{e}_{t+1} \\
=& \frac{1}{n}\sum_{i=1}^{n}\left(\frac{m_t^{(i)}}{b_t^{(i)}} + e_t^{(i)} - e_{t+1}^{(i)}\right) + \bar{e}_t - \bar{e}_{t+1} \\
=& \frac{1}{n}\sum_{i=1}^{n}\frac{m_t^{(i)}}{b_t^{(i)}} + \frac{1}{n}\sum_{i=1}^{n}\left(e_t^{(i)} - e_{t+1}^{(i)}\right) + \bar{e}_t - \bar{e}_{t+1}
\end{aligned}
\tag{7}
$$

Denote

$$
g_t \triangleq \frac{1}{n}\sum_{i=1}^{n} g_t^{(i)},
\tag{8}
$$

$$
m_t \triangleq \frac{1}{n}\sum_{i=1}^{n} m_t^{(i)} = \beta m_{t-1} + (1-\beta)g_t,
\tag{9}
$$

$$
b_t \triangleq \frac{1}{n}\sum_{i=1}^{n} b_t^{(i)},
\tag{10}
$$

$$
\delta_t \triangleq \frac{1}{n}\sum_{i=1}^{n}\frac{m_t^{(i)}}{b_t^{(i)}} - \frac{m_t}{b_t},
\tag{11}
$$

$$
e_t \triangleq \frac{1}{n}\sum_{i=1}^{n} e_t^{(i)} + \bar{e}_t
\tag{12}
$$

$$
\tag{13}
$$

Hence, the updating rule can be summarized as

$$
\begin{aligned}
x_{t+1} =& x_t - \alpha_t \bar{u}_t \\
=& x_t - \alpha_t \left(\frac{m_t}{b_t} + \delta_t + e_t - e_{t+1}\right)
\end{aligned}
\tag{14}
$$

### A.1   Auxiliary Lemmas

**Lemma 1.** *Let $u_t = \frac{m_t}{b_t}$, the element-wise quantization function is defined in Eq.(5) can be reformulated as*

$$
\mathcal{Q}\left((u_t)_j\right) = \left\{\begin{array}{ll} 1, & \text{with probability } p = \frac{(u_t)_j+1}{2} \\ -1, & \text{with probability } 1-p \end{array}\right. \quad (j \in \{1,2,...,d\}, \;\; -1 \le (u_t)_j \le 1).
\tag{15}
$$

*We have $e_t = u_t - \mathcal{Q}\left(u_t\right)$, and then the following holds true*

$$
\mathbb{E}\left[e_t\right] = 0, \;\; \mathbb{E}\left[\|e_t\|^2\right] \le d.
\tag{16}
$$

**Proof.** From Eq.(15), we know

$$
\begin{aligned}
\mathbb{E}\left[(e_t)_j\right] =& \mathbb{E}\left[u_t - \mathcal{Q}\left(u_t\right)\right] \\
=& \frac{1}{2}\left((u_t)_j + 1\right)\left((u_t)_j - 1\right) + (1 - \frac{1}{2}((u_t)_j+1))((u_t)_j + 1) = 0,
\end{aligned}
\tag{17}
$$

and,

$$\mathbb{E}\left[(e_t)_j^2\right] = \mathbb{E}\left[((u_t)_j - \mathcal{Q}((u_t)_j))^2\right]$$

$$= \frac{1}{2}((u_t)_j + 1)((u_t)_j - 1)^2 + (1 - \frac{1}{2}((u_t)_j + 1))((u_t)_j + 1)^2 \tag{18}$$

$$= 1 - ((u_t)_j)^2 \leq 1.$$

Hence,

$$\mathbb{E}[e_t] = 0, \ \mathbb{E}\left[\|e_t\|^2\right] \leq d. \tag{19}$$

***Lemma 2.*** *Let $x_0 = x_1$ and $\alpha_0 = \alpha_1$ in Algorithm 1, defining the sequence*

$$z_1 = x_1 + \alpha_1(\delta_1 - e_1) \tag{20}$$

$$z_t = x_t + \frac{\beta}{1 - \beta}(x_t - x_{t-1}) + \frac{\alpha_{t-1}}{1 - \beta}(\delta_{t-1} + \beta e_{t-1} - e_t), \forall t \geq 2. \tag{21}$$

*Then the following equality will hold, i.e.,*

$$z_{t+1} = z_t + \frac{\beta}{1 - \beta}\left(\frac{\alpha_{t-1}}{b_{t-1}} - \frac{\alpha_t}{b_t}\right) \odot m_{t-1} - \alpha_t \frac{g_t}{b_t} - \alpha_{t-1}\delta_{t-1} - (\alpha_t - \alpha_{t-1})e_t. \tag{22}$$

**Proof.** For $t = 1$, we have

$$z_2 - z_1 = x_2 + \frac{\beta}{1 - \beta}(x_2 - x_1) + \frac{\alpha_1}{1 - \beta}(\delta_1 + \beta e_1 - e_2) - (x_1 + \alpha_1(\delta_1 - e_1))$$

$$= (\frac{\beta}{1 - \beta} + 1)(x_2 - x_1) + \frac{\alpha_1}{1 - \beta}(\delta_1 + \beta e_1 - e_2) - \alpha_1(\delta_1 - e_1)$$

$$= -\frac{\alpha_1}{1 - \beta}\left(\frac{(1 - \beta)g_1}{b_1} + \delta_1 + e_1 - e_2\right) + \frac{\alpha_1}{1 - \beta}(\delta_1 + \beta e_1 - e_2) - \alpha_1(\delta_1 - e_1) \tag{23}$$

$$= -\alpha_1 \frac{g_1}{b_1} - \alpha_0\delta_1$$

where the second equality follows the updating rule in Eq.(14).

For $t \geq 2$, following the updating rule in Eq.(14), we have

$$x_{t+1} - x_t + \alpha_t(\delta_t + e_t - e_{t+1}) = -\alpha_t \frac{m_t}{b_t}$$

$$= -\alpha_t \frac{\beta m_{t-1} + (1 - \beta)g_t}{b_t}$$

$$= \beta(x_t - x_{t-1} + \alpha_{t-1}(\delta_t + e_{t-1} - e_t)) \tag{24}$$

$$+ \beta\left(\frac{\alpha_{t-1}}{b_{t-1}} - \frac{\alpha_t}{b_t}\right) \odot m_{t-1} - (1 - \beta)\alpha_t \frac{g_t}{b_t}$$

We know $x_{t+1} - x_t + \alpha_t(e_t - e_{t+1}) = (1 - \beta)(x_{t+1} + -\alpha_t(e_{t+1} - \delta_t)) - (1 - \beta)(x_t - \alpha_t e_t) + \beta(x_{t+1} - x_t + \alpha_t(\delta_t + e_t - e_{t+1}))$, so Eq. (24) can be rearranged as

$$(1 - \beta)(x_{t+1} + \alpha_t(\delta_t - e_{t+1})) + \beta(x_{t+1} - x_t + \alpha_t(\delta_t + e_t - e_{t+1}))$$

$$= (1 - \beta)(x_t - \alpha_t e_t) + \beta(x_t - x_{t-1} + \alpha_{t-1}(\delta_{t-1} + e_{t-1} - e_t)) \tag{25}$$

$$+ \beta\left(\frac{\alpha_{t-1}}{b_{t-1}} - \frac{\alpha_t}{b_t}\right) \odot m_{t-1} - (1 - \beta)\alpha_t \frac{g_t}{b_t}$$

Divided both sides by $1 - \beta$, we obtain

$$x_{t+1} + \alpha_t(\delta_t - e_{t+1}) + \frac{\beta}{1 - \beta}(x_{t+1} - x_t + \alpha_t(\delta_t + e_t - e_{t+1}))$$

$$= x_t + \alpha_{t-1}(\delta_{t-1} - e_t) + \frac{\beta}{1 - \beta}(x_t - x_{t-1} + \alpha_{t-1}(\delta_{t-1} + e_{t-1} - e_t))$$

$$+ \frac{\beta}{1 - \beta}\left(\frac{\alpha_{t-1}}{b_{t-1}} - \frac{\alpha_t}{b_t}\right) \odot m_{t-1} \tag{26}$$

$$- \alpha_t \frac{g_t}{b_t} - \alpha_{t-1}\delta_{t-1} - (\alpha_t - \alpha_{t-1})e_t$$

Rearranging Eq. (26), we have

$$
\begin{aligned}
& x_{t+1} + \frac{\beta}{1-\beta}(x_{t+1} - x_t) + \frac{\alpha_t}{1-\beta}(\delta_t + \beta e_t - e_{t+1}) \\
=& x_t + \frac{\beta}{1-\beta}(x_t - x_{t-1}) + \frac{\alpha_{t-1}}{1-\beta}(\delta_{t-1} + \beta e_{t-1} - e_t) \\
& + \frac{\beta}{1-\beta}\left(\frac{\alpha_{t-1}}{b_{t-1}} - \frac{\alpha_t}{b_t}\right) \odot m_{t-1} \\
& - \alpha_t \frac{g_t}{b_t} - \alpha_{t-1}\delta_{t-1} - (\alpha_t - \alpha_{t-1})e_t
\end{aligned}
\tag{27}
$$

Define the sequence

$$
z_t = x_t + \frac{\beta}{1-\beta}(x_t - x_{t-1}) + \frac{\alpha_{t-1}}{1-\beta}(\delta_{t-1} + \beta e_{t-1} - e_t)
\tag{28}
$$

We finally obtain

$$
z_{t+1} = z_t + \frac{\beta}{1-\beta}\left(\frac{\alpha_{t-1}}{b_{t-1}} - \frac{\alpha_t}{b_t}\right) \odot m_{t-1} - \alpha_t \frac{g_t}{b_t} - \alpha_{t-1}\delta_{t-1} - (\alpha_t - \alpha_{t-1})e_t.
\tag{29}
$$

Recalling $x_1 = x_0$ and $\alpha_1 = \alpha_0$, we have $\frac{\alpha_1}{b_1} = \frac{\alpha_0}{b_0}$. Then, combining Eq.(23) and Eq.(29), we obtain the conclusion.

## A.2 Proof of Theorem 1

**Proof.** By the the gradient Lipschitz continuous in Assumption 2 and Lemma 2, we obtain

$$
\begin{aligned}
\mathbb{E}[f(z_{t+1}) - f(z_t)] \leq & \; \mathbb{E}\langle \nabla f(z_t), z_{t+1} - z_t \rangle + \frac{L}{2}\mathbb{E}\|z_{t+1} - z_t\|^2 \\
=& \mathbb{E}\left[\frac{\beta}{1-\beta}\langle \nabla f(z_t), \left(\frac{\alpha_{t-1}}{b_{t-1}} - \frac{\alpha_t}{b_t}\right) \odot m_{t-1}\rangle\right] - \mathbb{E}\left[\langle \nabla f(z_t), \alpha_t \frac{g_t}{b_t}\rangle\right] \\
& - \mathbb{E}\left[\langle \nabla f(z_t), \alpha_{t-1}\delta_{t-1}\rangle\right] - \mathbb{E}\left[\langle \nabla f(z_t), (\alpha_t - \alpha_t)e_{t-1}\rangle\right] \\
& + \mathbb{E}\left[\frac{L}{2}\left\|\frac{\beta}{1-\beta}\left(\frac{\alpha_{t-1}}{b_{t-1}} - \frac{\alpha_t}{b_t}\right) \odot m_{t-1} - \alpha_t \frac{g_t}{b_t} - \alpha_{t-1}\delta_{t-1} - (\alpha_t - \alpha_{t-1})e_{t-1}\right\|^2\right] \\
=& \mathbb{E}\left[\frac{\beta}{1-\beta}\langle \nabla f(z_t), \left(\frac{\alpha_{t-1}}{b_{t-1}} - \frac{\alpha_t}{b_t}\right) \odot m_{t-1}\rangle\right] - \mathbb{E}\left[\langle \nabla f(z_t), \alpha_t \frac{g_t}{b_t}\rangle\right] \\
& + \mathbb{E}\left[\frac{L}{2}\left\|\frac{\beta}{1-\beta}\left(\frac{\alpha_{t-1}}{b_{t-1}} - \frac{\alpha_t}{b_t}\right) \odot m_{t-1} - \alpha_t \frac{g_t}{b_t} - \alpha_{t-1}\delta_{t-1} - (\alpha_t - \alpha_{t-1})e_{t-1}\right\|^2\right] \\
\leq& \mathbb{E}\left[\frac{\beta}{1-\beta}\langle \nabla f(z_t), \left(\frac{\alpha_{t-1}}{b_{t-1}} - \frac{\alpha_t}{b_t}\right) \odot m_{t-1}\rangle\right] - \mathbb{E}\left[\langle \nabla f(z_t), \alpha_t \frac{g_t}{b_t}\rangle\right] \\
& + L\mathbb{E}\left[\left\|\frac{\beta}{1-\beta}\left(\frac{\alpha_{t-1}}{b_{t-1}} - \frac{\alpha_t}{b_t}\right) \odot m_{t-1}\right\|^2\right] + L\mathbb{E}\left[\alpha_t^2 \left\|\frac{g_t}{b_t}\right\|^2\right] \\
& + \frac{L}{2}\mathbb{E}\left[\|\alpha_{t-1}\delta_{t-1}\|^2\right] + \frac{L}{2}\mathbb{E}\left[\|(\alpha_{t-1} - \alpha_t)e_t\|^2\right]
\end{aligned}
\tag{30}
$$

where the second equality holds due to $\mathbb{E}\left[\delta_{t-1}\right] = 0$ and $\mathbb{E}\left[e_{t-1}\right] = 0$. The last inequality holds owing to $\mathbb{E}[\|a + b\|^2] = \mathbb{E}[\|a\|^2] + \mathbb{E}[\|b\|^2]$ if $\mathbb{E}[a] = 0$ or $\mathbb{E}[b] = 0$, and $\mathbb{E}[\|a + b\|^2] \leq 2\mathbb{E}[\|a\|^2] + 2\mathbb{E}[\|b\|^2]$ if $\mathbb{E}[a] \neq 0$ and $\mathbb{E}[b] \neq 0$.

Taking telescope sum from 1 to $T$ on the both sides of Eq.(30) , we then have

$$
\mathbb{E}[f(z_T) - f(z_1)] \leq \underbrace{\frac{\beta}{1-\beta}\mathbb{E}\left[\sum_{t=1}^{T}\langle\nabla f(z_t), \left(\frac{\alpha_{t-1}}{b_{t-1}} - \frac{\alpha_t}{b_t}\right)\odot m_{t-1}\rangle\right]}_{T_1} \underbrace{-\mathbb{E}\left[\sum_{t=1}^{T}\langle\nabla f(z_t), \alpha_t\frac{g_t}{b_t}\rangle\right]}_{T_2}
$$

$$
+ \underbrace{L\mathbb{E}\left[\sum_{t=1}^{T}\left\|\frac{\beta}{1-\beta}\left(\frac{\alpha_{t-1}}{b_{t-1}} - \frac{\alpha_t}{b_t}\right)\odot m_{t-1}\right\|^2\right]}_{T_3}
$$

$$
+ \underbrace{L\mathbb{E}\left[\sum_{t=1}^{T}\alpha_t^2\left\|\frac{g_t}{b_t}\right\|^2\right] + \frac{L}{2}\mathbb{E}[\sum_{t=1}^{T}\|\alpha_{t-1}\delta_{t-1}\|^2] + \frac{L}{2}\mathbb{E}[\sum_{t=1}^{T}\|(\alpha_{t-1} - \alpha_t)e_t\|^2]}_{T_4}
$$

$$(31)$$

Now we focus on bounding $T_1$ below. From Assumption 4, we know $\|g_t\| \leq G$ ($t = 1, 2, ..., T$) and $\|\nabla f(z_t)\| \leq G$ . Due to $m_t = \beta m_{t-1} + (1-\beta)g_t$ and $m_1 = g_1$, it is easy to obtain $\|m_t\| \leq G$ by complete induction.

Since $\|\nabla f(z_t)\| \leq G$ and $\|m_t\| \leq G$, we have

$$
\begin{aligned}
T_1 &= \frac{\beta}{1-\beta}\mathbb{E}\left[\sum_{i=1}^{T}\langle\nabla f(z_i), \left(\frac{\alpha_{t-1}}{b_{t-1}} - \frac{\alpha_t}{b_t}\right)\odot m_{i-1}\rangle\right] \\
&\overset{(i)}{\leq} \frac{\beta}{1-\beta}\mathbb{E}\left[\sum_{i=1}^{T}\|\nabla f(z_t)\|\|m_t\|\left\|\frac{\alpha_{t-1}}{b_{t-1}} - \frac{\alpha_t}{b_t}\right\|_1\right] \\
&\overset{(ii)}{\leq} \frac{\beta}{1-\beta}G^2\mathbb{E}\left[\sum_{i=1}^{T}\left\|\frac{\alpha_{t-1}}{b_{t-1}} - \frac{\alpha_t}{b_t}\right\|_1\right] \\
&\overset{(iii)}{=} \frac{\beta}{1-\beta}G^2\mathbb{E}\left[\left\|\sum_{i=1}^{T}\left(\frac{\alpha_{t-1}}{b_{t-1}} - \frac{\alpha_t}{b_t}\right)\right\|_1\right] \\
&\leq \frac{\beta}{1-\beta}G^2\mathbb{E}\left[\left\|\frac{\alpha_0}{b_0}\right\|_1\right] \\
&\overset{(iv)}{\leq} \frac{\alpha_0\beta d}{(1-\beta)\rho}G^2,
\end{aligned}
$$

$$(32)$$

where $(i)$ holds sice $\|a\odot b\| \leq \|a\|\max_j|(b)_j| \leq \|a\|\|b\|_1$, $(ii)$ holds due to $\|\nabla f(z_t)\| \leq G$ and $\|m_t\| \leq G$, $(iii)$ holds because $\frac{\alpha_{t-1}}{(b_{t-1})_j} - \frac{\alpha_t}{(b_t)_j} \geq 0$ for any $j \in [1, 2, ..., d]$, $(iv)$ holds due to $\min_j(b_t)_j \geq \rho > 0$ for any $j \in [1, 2, ..., d]$.

Let us turn to bound $T_2$,

$$
\begin{aligned}
T_2 &= -\mathbb{E}\left[\sum_{t=1}^{T}\langle\nabla f(z_t), \alpha_t\frac{g_t}{b_t}\rangle\right] \\
&= \underbrace{-\mathbb{E}\left[\sum_{t=1}^{T}\langle\nabla f(z_t) - f(x_t), \alpha_t\frac{g_t}{b_t}\rangle\right]}_{T_5} \underbrace{-\mathbb{E}\left[\sum_{t=1}^{T}\langle\nabla f(x_t), \alpha_t\frac{g_t}{b_t}\rangle\right]}_{T_6}
\end{aligned}
$$

$$(33)$$

We now analyze $T_5$ below,

$$T_5 = -\mathbb{E}\left[\sum_{t=1}^{T}\langle\nabla f(z_t) - f(x_t), \alpha_t\frac{g_t}{b_t}\rangle\right]$$

$$\overset{(i)}{\leq} \frac{1}{2}\mathbb{E}\left[\sum_{t=1}^{T}\|f(z_t) - f(x_t)\|^2\right] + \frac{1}{2}\mathbb{E}\left[\sum_{t=1}^{T}\alpha_t^2\left\|\frac{g_t}{b_t}\right\|^2\right]$$

$$\overset{(ii)}{\leq} \frac{L^2}{2}\mathbb{E}\left[\sum_{t=1}^{T}\|z_t - x_t\|^2\right] + \frac{1}{2}\mathbb{E}\left[\sum_{t=1}^{T}\alpha_t^2\left\|\frac{g_t}{b_t}\right\|^2\right]$$

$$\overset{(iii)}{=} \frac{L^2}{2}\mathbb{E}\left[\sum_{t=1}^{T}\left\|\frac{\beta}{1-\beta}(x_t - x_{t-1}) + \frac{\alpha_{t-1}}{1-\beta}\left(\delta_{t-1} + \beta e_{t-1} - e_t\right)\right\|^2\right] + \frac{1}{2}\mathbb{E}\left[\sum_{t=1}^{T}\alpha_t^2\left\|\frac{g_t}{b_t}\right\|^2\right]$$

$$\overset{(iv)}{\leq} \frac{\beta^2 L^2}{(1-\beta)^2}\mathbb{E}\left[\sum_{t=1}^{T}\|x_t - x_{t-1}\|^2\right] + \frac{L^2}{(1-\beta)^2}\mathbb{E}\left[\sum_{t=1}^{T}\|\alpha_{t-1}\delta_{t-1}\|^2\right]$$

$$+ \frac{\beta^2 L^2}{(1-\beta)^2}\mathbb{E}\left[\sum_{t=1}^{T}\|\alpha_{t-1}e_{t-1}\|^2\right] + \frac{L^2}{(1-\beta)^2}\mathbb{E}\left[\sum_{t=1}^{T}\|\alpha_{t-1}e_t\|^2\right] + \frac{1}{2}\mathbb{E}\left[\sum_{t=1}^{T}\alpha_t^2\left\|\frac{g_t}{b_t}\right\|^2\right]$$

$$\overset{(v)}{=} \frac{\beta^2 L^2}{(1-\beta)^2}\mathbb{E}\left[\sum_{t=1}^{T}\alpha_{t-1}^2\left\|\left(\frac{m_{t-1}}{b_{t-1}} + \delta_{t-1} + e_{t-1} - e_t\right)\right\|^2\right] + \frac{L^2}{(1-\beta)^2}\mathbb{E}\left[\sum_{t=1}^{T}\|\alpha_{t-1}\delta_{t-1}\|^2\right]$$

$$+ \frac{\beta^2 L^2}{(1-\beta)^2}\mathbb{E}\left[\sum_{t=1}^{T}\|\alpha_{t-1}e_{t-1}\|^2\right] + \frac{L^2}{(1-\beta)^2}\mathbb{E}\left[\sum_{t=1}^{T}\|\alpha_{t-1}e_t\|^2\right] + \frac{1}{2}\mathbb{E}\left[\sum_{t=1}^{T}\alpha_t^2\left\|\frac{g_t}{b_t}\right\|^2\right]$$

$$= \frac{\beta^2 L^2}{(1-\beta)^2}\mathbb{E}\left[\sum_{t=1}^{T}\left\|\frac{\alpha_{t-1}m_{t-1}}{b_{t-1}}\right\|^2\right] + \frac{\beta^2 L^2}{(1-\beta)^2}\mathbb{E}\left[\sum_{t=1}^{T}\|\alpha_{t-1}\delta_{t-1}\|^2\right]$$

$$+ \frac{\beta^2 L^2}{(1-\beta)^2}\mathbb{E}\left[\sum_{t=1}^{T}\|\alpha_{t-1}e_{t-1}\|^2\right] + \frac{\beta^2 L^2}{(1-\beta)^2}\mathbb{E}\left[\sum_{t=1}^{T}\|\alpha_{t-1}e_t\|^2\right] + \frac{L^2}{(1-\beta)^2}\mathbb{E}\left[\sum_{t=1}^{T}\|\alpha_{t-1}\delta_{t-1}\|^2\right]$$

$$+ \frac{\beta^2 L^2}{(1-\beta)^2}\mathbb{E}\left[\sum_{t=1}^{T}\|\alpha_{t-1}e_{t-1}\|^2\right] + \frac{L^2}{(1-\beta)^2}\mathbb{E}\left[\sum_{t=1}^{T}\|\alpha_{t-1}e_t\|^2\right] + \frac{1}{2}\mathbb{E}\left[\sum_{t=1}^{T}\alpha_t^2\left\|\frac{g_t}{b_t}\right\|^2\right]$$

$$= \frac{\beta^2 L^2}{(1-\beta)^2}\mathbb{E}\left[\sum_{t=1}^{T}\alpha_{t-1}^2\left\|\frac{m_{t-1}}{b_{t-1}}\right\|^2\right] + \frac{(1+\beta^2)L^2}{(1-\beta)^2}\mathbb{E}\left[\sum_{t=1}^{T}\alpha_{t-1}^2\|\delta_{t-1}\|^2\right]$$

$$+ \frac{2\beta^2 L^2}{(1-\beta)^2}\mathbb{E}\left[\sum_{t=1}^{T}\alpha_{t-1}^2\|e_{t-1}\|^2\right] + \frac{(1+\beta^2)L^2}{(1-\beta)^2}\mathbb{E}\left[\sum_{t=1}^{T}\alpha_{t-1}^2\|e_t\|^2\right] + \frac{1}{2}\mathbb{E}\left[\sum_{t=1}^{T}\alpha_t^2\left\|\frac{g_t}{b_t}\right\|^2\right]$$

$$\overset{(vi)}{\leq} \left(\frac{\beta^2 L^2 d}{(1-\beta)^2} + \frac{4(1+\beta^2)L^2 d}{(1-\beta)^2} + \frac{2\beta^2 L^2 d}{(1-\beta)^2} + \frac{(1+\beta^2)L^2 d}{(1-\beta)^2} + \frac{G^2}{2\rho^2}\right)\sum_{t=1}^{T}\alpha_{t-1}^2$$

$$= \left(\frac{(8\beta^2 + 10\beta + 5)L^2 d}{(1-\beta)^2} + \frac{G^2}{2\rho^2}\right)\sum_{t=1}^{T}\alpha_{t-1}^2$$

(34)

where $(i)$ holds by following $\langle a,b\rangle \leq \frac{1}{2}\|a\|^2 + \frac{1}{2}\|a\|^2$, $(ii)$ holds due to Assumption 1, $(iii)$ holds due to Assumption 1 owing to Eq.(21), $(iii)$ holds since $\mathbb{E}[\|a + b\|^2] = \mathbb{E}[\|a\|^2] + \mathbb{E}[\|b\|^2]$ if $\mathbb{E}[a] = 0$ or $\mathbb{E}[b] = 0$, $(v)$ holds resulting from the updating rule in Eq. (14), $(vi)$ holds due to $\left|\frac{(m_t)_j}{(b_t)_j}\right| \leq 1$, $|(\delta)_j| \leq 2$ (the definition of $\delta_t$ in Eq. (11) ), $\mathbb{E}[\|e_t\|^2] \leq d$ in Lemma 1, $\|g_t\| \leq G$ in Assumption 2 and $\min_j(b_t)_j \geq \rho > 0$.

We then bound $T_6$

$$
\begin{aligned}
T_6 = & -\mathbb{E}\left[\sum_{t=1}^{T}\langle\nabla f(x_t), \alpha_t\frac{g_t}{b_t}\rangle\right]\\
= & -\mathbb{E}\left[\sum_{t=1}^{T}\langle\nabla f(x_t), \alpha_t\frac{\nabla f(x_t)}{b_t}\rangle\right] - \mathbb{E}\left[\sum_{t=1}^{T}\langle\nabla f(x_t), \alpha_t\frac{g_t - \nabla f(x_t)}{b_t}\rangle\right]\\
\overset{(i)}{\leq} & -\frac{1}{G}\mathbb{E}\left[\sum_{t=1}^{T}\alpha_t\|\nabla f(x_t)\|^2\right] + \mathbb{E}\left[\sum_{t=1}^{T}\langle\nabla f(x_t), \alpha_t\frac{\nabla f(x_t) - g_t}{b_t}\rangle\right]\\
= & -\frac{1}{G}\mathbb{E}\left[\sum_{t=1}^{T}\alpha_t\|\nabla f(x_t)\|^2\right] + \mathbb{E}\left[\langle\nabla f(x_1), \alpha_1\frac{\nabla f(x_1) - g_1}{b_1}\rangle\right]\\
& + \mathbb{E}\left[\sum_{t=2}^{T}\langle\nabla f(x_t), \nabla(f(x_t) - g_t)\odot\left(\frac{\alpha_t}{b_t} - \frac{\alpha_{t-1}}{b_{t-1}}\right)\rangle\right] + \mathbb{E}\left[\sum_{t=2}^{T}\langle\nabla f(x_t), \alpha_{t-1}\frac{\nabla f(x_t) - g_t}{b_{t-1}}\rangle\right]\\
\overset{(ii)}{=} & -\frac{1}{G}\mathbb{E}\left[\sum_{t=1}^{T}\alpha_t\|\nabla f(x_t)\|^2\right] + \mathbb{E}\left[\langle\nabla f(x_1), \alpha_1\frac{\nabla f(x_1) - g_1}{b_1}\rangle\right]\\
& + \mathbb{E}\left[\sum_{t=2}^{T}\langle\nabla f(x_t), (\nabla f(x_t) - g_t)\odot\left(\frac{\alpha_t}{b_t} - \frac{\alpha_{t-1}}{b_{t-1}}\right)\rangle\right]\\
\overset{(iii)}{\leq} & -\frac{1}{G}\mathbb{E}\left[\sum_{t=1}^{T}\alpha_t\|\nabla f(x_t)\|^2\right] + \mathbb{E}\left[\|\nabla f(x_1)\|\|\nabla f(x_1) - g_1\|\left\|\frac{\alpha_1}{b_1}\right\|_1\right]\\
& + \mathbb{E}\left[\sum_{t=2}^{T}\|\nabla f(x_t)\|\|\nabla f(x_t) - g_t\|\left\|\frac{\alpha_t}{b_t} - \frac{\alpha_{t-1}}{b_{t-1}}\right\|_1\right]\\
\overset{(iv)}{\leq} & -\frac{1}{G}\mathbb{E}\left[\sum_{t=1}^{T}\alpha_t\|\nabla f(x_t)\|^2\right] + 2G^2\mathbb{E}\left[\left\|\frac{\alpha_1}{b_1}\right\|_1 + \sum_{t=2}^{T}\left\|\frac{\alpha_t}{b_t} - \frac{\alpha_{t-1}}{b_{t-1}}\right\|_1\right]\\
\overset{(v)}{=} & -\frac{1}{G}\mathbb{E}\left[\sum_{t=1}^{T}\alpha_t\|\nabla f(x_t)\|^2\right] + 2G^2\mathbb{E}\left[\left\|\frac{\alpha_1}{b_1} + \sum_{t=2}^{T}\frac{\alpha_{t-1}}{b_{t-1}} - \frac{\alpha_t}{b_t}\right\|_1\right],\\
= & -\frac{1}{G}\mathbb{E}\left[\sum_{t=1}^{T}\alpha_t\|\nabla f(x_t)\|^2\right] + 4G^2\mathbb{E}\left[\left\|\frac{\alpha_1}{b_1}\right\|_1\right],\\
\overset{(vi)}{\leq} & -\frac{1}{G}\mathbb{E}\left[\sum_{t=1}^{T}\alpha_t\|\nabla f(x_t)\|^2\right] + \frac{4G^2\alpha_1 d}{\rho}
\end{aligned}
$$

(35)

where $(i)$ holds due to $\max_j(b_t)_j \leq \|b_t\| \leq G$ , $(ii)$ holds owing to $\mathbb{E}[\nabla f(x_t) - g_t] = 0$ in Assumption 2 and $g_t, b_{t-1}$ are independent, $(iii)$ holds sice $\|a\odot b\| \leq \|a\|\max_j|(b)_j| \leq \|a\|\|b\|_1$, $(iv)$ holds resulting from $\|\nabla f(x_t)\| \leq G$ and $\|\nabla f(x_t) - g_t\| \leq \|\nabla f(x_t)\| + \|g_t\| \leq 2G$, and $(v)$ holds because $\frac{\alpha_{t-1}}{(b_{t-1})_j} - \frac{\alpha_t}{(b_t)_j} \geq 0$ for any $j \in [1, 2, ..., d]$, $(vi)$ holds due to $\min_j(b_t)_j \geq \rho > 0$ for any $j \in [1, 2, ..., d]$.

Then, we pay attention to $T_3$,

$$
\begin{aligned}
T_3 &= L\mathbb{E}\left[\sum_{t=1}^{T}\left\|\frac{\beta}{1-\beta}\left(\frac{\alpha_{t-1}}{b_{t-1}}-\frac{\alpha_t}{b_t}\right)\odot m_{t-1}\right\|^2\right]\\
&\overset{(i)}{\leq}\frac{\beta^2 L}{(1-\beta)^2}\mathbb{E}\left[\sum_{t=1}^{T}\left\|\frac{\alpha_{t-1}}{b_{t-1}}-\frac{\alpha_t}{b_t}\right\|^2\|m_{t-1}\|^2\right]\\
&\overset{(ii)}{\leq}\frac{\beta^2 LG^2}{(1-\beta)^2}\mathbb{E}\left[\sum_{t=1}^{T}\left\|\frac{\alpha_{t-1}}{b_{t-1}}-\frac{\alpha_t}{b_t}\right\|^2\right]\\
&\overset{(iii)}{\leq}\frac{\beta^2 LG^2}{(1-\beta)^2}\mathbb{E}\left[\sum_{t=1}^{T}\max_j\left|\frac{\alpha_{t-1}}{(b_{t-1})_j}-\frac{\alpha_t}{(b_t)_j}\right|\left\|\frac{\alpha_{t-1}}{b_{t-1}}-\frac{\alpha_t}{b_t}\right\|_1\right]\\
&\overset{(iv)}{\leq}\frac{\alpha_0\beta^2 LG^2}{\rho(1-\beta)^2}\mathbb{E}\left[\sum_{t=1}^{T}\max_j\left(\frac{\alpha_{t-1}}{(b_{t-1})_j}\right)\left\|\frac{\alpha_{t-1}}{b_{t-1}}-\frac{\alpha_t}{b_t}\right\|_1\right]\\
&\overset{(v)}{\leq}\frac{\alpha_0\beta^2 LG^2}{\rho(1-\beta)^2}\mathbb{E}\left[\sum_{t=1}^{T}\left\|\frac{\alpha_{t-1}}{b_{t-1}}-\frac{\alpha_t}{b_t}\right\|_1\right]\\
&\overset{(vi)}{\leq}\frac{\alpha_0\beta^2 LG^2}{\rho(1-\beta)^2}\mathbb{E}\left[\sum_{t=1}^{T}\left\|\frac{\alpha_{t-1}}{b_{t-1}}\right\|_1-\left\|\frac{\alpha_t}{b_t}\right\|_1\right]\\
&\overset{(vii)}{\leq}\frac{\alpha_0\beta^2 LG^2}{\rho(1-\beta)^2}\mathbb{E}\left[\left\|\frac{\alpha_0}{b_0}\right\|_1-\left\|\frac{\alpha_T}{b_T}\right\|_1\right]\\
&\overset{(viii)}{\leq}\frac{\alpha_0^2\beta^2 LG^2 d}{\rho^2(1-\beta)^2},
\end{aligned}
\tag{36}
$$

where $(i)$ holds due to $\|a\odot b\|\leq\|a\|\|b\|$, $(ii)$ holds owing to $\|m_{t-1}\|\leq G$, $(ii)$ holds due to $\|a\|^2\leq\max_j|(a)_j|\|a\|_1$ , $(iv)$ holds due to $\frac{\alpha_{t-1}}{(b_{t-1})_j}-\frac{\alpha_t}{(b_t)_j}\geq 0$ and $\frac{\alpha_t}{(b_t)_j}>0$ for any $j\in[1,2,...,d]$, $(v)$ holds resulting from $\min_j(b_t)_j\geq\rho>0$ for any $j$ and $\alpha_t$ is non-increasing, $(vi)$ holds resulting from $\frac{\alpha_{t-1}}{(b_{t-1})_j}-\frac{\alpha_t}{(b_t)_j}\geq 0$ for any $j\in[1,2,...,d]$, $(vii)$ holds due to telescoping sum, and $(viii)$ holds due to $\min_j(b_t)_j\geq\rho>0$ for any $j\in[1,2,...,d]$..

Now we turn attention to $T_4$,

$$
\begin{aligned}
T_4 &= L\mathbb{E}\left[\sum_{t=1}^{T}\alpha_t^2\left\|\frac{g_t}{b_t}\right\|^2\right]+\frac{L}{2}\mathbb{E}[\sum_{t=1}^{T}\|\alpha_{t-1}\delta_{t-1}\|^2]+\frac{L}{2}\mathbb{E}[\sum_{t=1}^{T}\|(\alpha_{t-1}-\alpha_t)e_t\|^2]\\
&\leq\left(L\frac{G^2}{\rho^2}+2dL\right)\sum_{t=1}^{T}\alpha_t^2+\frac{dL}{2}\sum_{t=1}^{T}(\alpha_{t-1}-\alpha_t)^2,
\end{aligned}
\tag{37}
$$

where the inequality holds owing to $\|m_{t-1}\|\leq G$ and $\min_j(b_t)_j\geq\rho>0$, $\|(\delta_{t-1})_j\|\leq 2$, and $\mathbb{E}[\|e_t\|^2]\leq d$.

Combining Eq.(31-37), we can obtain

$$
\begin{aligned}
\mathbb{E}[f(z_T)-f(z_1)]\leq{}&\frac{\alpha_0\beta d}{(1-\beta)\rho}G^2+\left(\frac{(8\beta^2+10\beta+5)L^2 d}{(1-\beta)^2}+\frac{G^2}{2\rho^2}\right)\sum_{t=1}^{T}\alpha_{t-1}^2\\
&-\frac{1}{G}\mathbb{E}\left[\sum_{t=1}^{T}\alpha_t\|\nabla f(x_t)\|^2\right]+\frac{4G^2\alpha_1 d}{\rho}+\frac{\alpha_0^2\beta^2 LG^2 d}{\rho^2(1-\beta)^2}\\
&+\left(L\frac{G^2}{\rho^2}+2dL\right)\sum_{t=1}^{T}\alpha_t^2+\frac{dL}{2}\sum_{t=1}^{T}(\alpha_{t-1}-\alpha_t)^2.
\end{aligned}
\tag{38}
$$

Reformulating Eq.(38), we then have

$$\frac{1}{G}\mathbb{E}\left[\sum_{t=1}^{T}\alpha_t\|\nabla f(x_t)\|^2\right] \leq \mathbb{E}[f(z_1)-f(z_T)]$$

$$+ \left(\frac{(8\beta^2+10\beta+5)L^2d}{(1-\beta)^2}+\frac{G^2(1+L)}{2\rho^2}+2dL\right)\sum_{t=1}^{T}\alpha_{t-1}^2$$

$$+ \frac{dL}{2}\sum_{t=1}^{T}(\alpha_{t-1}-\alpha_t)^2$$

$$+ \frac{\alpha_0\beta d}{(1-\beta)\rho}G^2 + \frac{4G^2\alpha_1 d}{\rho} + \frac{\alpha_0^2\beta^2 LG^2 d}{\rho^2(1-\beta)^2} \tag{39}$$

It is known the learning rate saftifies $\alpha_t = \frac{c}{\sqrt{t}}, \forall t \geq 1$ and $\alpha_0 = \alpha_1 = c$. Utilizing non-increasing $\alpha_t$ and Cauchy-Schwarz inequality, we know $\mathbb{E}\left[\sum_{t=1}^{T}\alpha_t\|\nabla f(x_t)\|^2\right] \geq T\alpha_T\mathbb{E}\left[\frac{1}{T}\sum_{t=1}^{T}\|\nabla f(x_t)\|\right]^2 = \frac{\sqrt{T}}{c}\mathbb{E}\left[\frac{1}{T}\sum_{t=1}^{T}\|\nabla f(x_t)\|\right]^2$. $\sum_{t=1}^{T}\alpha_{t-1}^2 = \sum_{t=1}^{T}\frac{c^2}{t} \leq c^2(1+\int_{1}^{T-1}\frac{1}{t}dt) \leq c^2(1+\log T)$, and $\sum_{t=1}^{T}(\alpha_{t-1}-\alpha_t)^2 = \sum_{t=2}^{T}(\alpha_{t-1}-\alpha_t)^2 \leq \sum_{t=2}^{T}\frac{c^2}{4(t-1)^3} \leq \frac{c^2}{4}(1+\int_{1}^{T-2}t^{-3}dt) = \frac{c^2}{4}(\frac{3}{2}-\frac{1}{2(T-2)}) \leq \frac{3c^2}{8}$, we further have

$$\mathbb{E}\left[\frac{1}{T}\sum_{t=1}^{T}\|\nabla f(x_t)\|\right]^2 \leq \frac{C_1}{\sqrt{T}}+\frac{C_2(1+\log T)}{\sqrt{T}}, \tag{40}$$

where we define

$$C_1 = cG\left(\mathbb{E}[f(z_1)-f^*]+\frac{3c^2dL}{16}+\frac{\beta cdG^2}{(1-\beta)\rho}+\frac{4cdG^2}{\rho}+\frac{c^2\beta^2 LG^2 d}{\rho^2(1-\beta)^2}\right), \tag{41}$$

$$C_2 = c^3 G\left(\frac{(8\beta^2+10\beta+5)L^2 d}{(1-\beta)^2}+\frac{G^2(1+L)}{2\rho^2}+2dL\right). \tag{42}$$

# B   Experiments for Comparing Vanilla SGD, SGDM, Adam, Birder and SoftSignSGD

To address the bottleneck in communication during distributed training, numerous gradient compression algorithms have been proposed, aiming to reduce the communication volume. Most of these algorithms can be reduced to Vanilla SGD without momentum if compression is not performed. Generally speaking, the epoch-wise convergence rate and inference performance a compressed algorithms is upper bounded by its uncompressed counterpart. In the experiments, we conducted empirical experiments to evaluate the training and inference performance of of Vanilla SGD, SGDM, Adam, Birder and its uncompressed version in training typical CNN-base, LSTM-base and Transformer-base DNNs.

---

**Algorithm 1.** SoftSignSGD

1: **Input**: model parameter $x_0, x_1$ , the momentum $m_0^{(i)} = 0$, $b_0^{(i)} = 0$, the exponential moving average factor $\beta$, the learning rate sequence $\{\alpha_t\}$
2: **for** $t = 1, ..., T$ **do**
3:     Randomly sample $\xi_t$ and compute the gradient: $g_t = \nabla f(x_t; \xi_t)$
4:     Update the momentum $m_t$: $m_t = \beta m_{t-1} + (1-\beta)g_t$
5:     Update the momentum $b_t$: $b_t = \beta b_{t-1} + (1-\beta)|g_t|$
6:     Update the model parameter $x_{t+1}$: $x_{t+1} = x_t - \alpha_t\frac{m_t}{b_t}$
7: **end for**

---

We refer to the uncompressed Birder as SoftSignSGD. The implementation details for SoftSignSGD are presented in Algorithm 1. When comparing SoftSignSGD to Adam, there are two key differences. First, instead of using the square root of the exponential moving average of the squared gradient, denoted as $\sqrt{v_t} = \sqrt{(1-\beta_2)v_{t-1}+(1-\beta_2)g_t^2}$, SoftSignSGD utilizes the exponential moving average of the absolute gradient, represented as $b_t = (1-\beta)b_{t-1}+|g_t|$. Second, in SoftSignSGD, the exponential moving factors for both the numerator $m_t$ and the denominator $b_t$ are the same. These differences ensure that each element of the updating amount in SoftSignSGD satisfies the condition $-1 \leq (\frac{m_t}{b_t})_j \leq 1$.

## B.1 Experimental Results for training ResNet-20

We evaluated the performance of five optimization algorithms: Vanilla SGD, SGDM, AdamW, Birder and SoftSignSGD, for training ResNet-20 on CIFAR100. Each batch consisted of a set of 128 examples sampled with replacement. For SGDM, we set the momentum parameter $\beta$ to 0.9, while for SoftSignSGD and Birder, it was set to 0.95. For AdamW, the parameters $\beta_1$ and $\beta_2$ were set to 0.9 and 0.999, respectively. The weight decay was uniformly set to 0.0005 for Vanilla SGD and SGDM, and 0.05 for AdamW, Birder and SoftSignSGD. To simplify the tuning process and ensure fair comparisons, we initialized the learning rates at 0.1 for Vanilla SGD and SGDM, and 0.005 for AdamW, Birder and SoftSignSGD. We divided the learning rates by 10 after 75 and 130 epochs, and terminated the training after 150 epochs.

Figure 5 visually demonstrates that Vanilla SGD exhibits slower convergence speed and lower test accuracy compared to SGDM. In contrast, both Birder and SoftSignSGD show comparable training and inference performance to the commonly used SGDM and AdamW. This observation suggests that Birder outperforms existing gradient compression algorithms when training CNN-based ResNet-20 models.

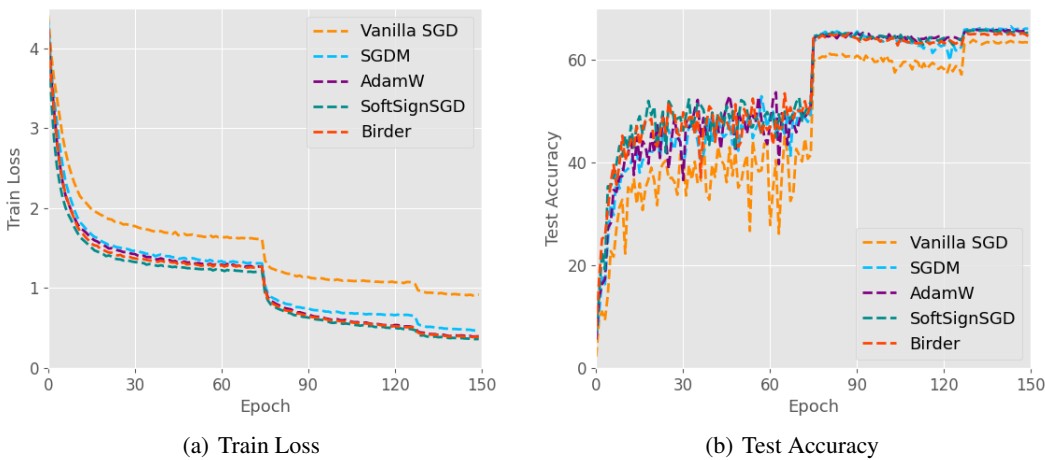

(a) Train Loss · · · · · · · · · · · · · · · · · · · · · (b) Test Accuracy

Figure 5: Training loss and test accuracy for ResNet-20 on CIFAR100.

## B.2 Experimental Results for training LSTM

We conducted experiments to train a 3-layer LSTM model on the Penn TreeBank dataset to evaluate the performance of five optimization algorithms: Vanilla SGD, SGDM, AdamW, Birder and SoftSignSGD. Our implementations were built upon the code provided in the AdaBelief paper[7], and we used the default experimental settings for SGDM and AdamW. For Vanilla SGD, we used the experimental settings of SGDM with the exception that we set the momentum parameter $\beta$ to 0. For Birder and SoftSignSGD, we adopted the experimental settings of AdamW, except that we set the momentum parameter $\beta$ to 0.99.

As visually illustrated in Figure 6, Vanilla SGD is still less effective than SGDM in terms of the convergence speed and the test accuracy, while the training and inference performance of SoftSignSGD and Birder are comparative to common-used SGDM and AdamW. It indicates the Birder is superior to exiting gradient compression algorithms for training LSTM.

## B.3 Experimental Results for training ViT

We train ViT-B with Vanilla SGD, SGDM, AdamW, SoftSignSGD and Birder on the ILSVRC2012 with 32 GPUs (4 nodes). We use the Pytorch official implementation for ViT [8]. For AdamW, SoftSignSGD and Birder, we followed the recommended experimental settings, with the exception that we set the momentum parameter $\beta$ to 0.95 for SoftSignSGD and Birder. As for Vanilla SGD and SGDM, we set the basic learning rate to 0.1 and the weight decay to 0.001, while keeping other settings the same as AdamW. Instead of the default 300 epochs, we uniformly set the total number of epochs to 150 for all optimizers

As visually illustrated in Figure 7, Vanilla SGD is still less effective than SGDM in terms of the convergence speed and the test accuracy, while the training and inference performance of SoftSignSGD and Birder are comparative to common-used SGDM and AdamW. Notably, the performance of SGD-type optimizers are substantially inferior to that of adaptive optimizers.

---

[7]https://github.com/juntang-zhuang/Adabelief-Optimizer

[8]https://github.com/pytorch/vision/tree/main/references/classification

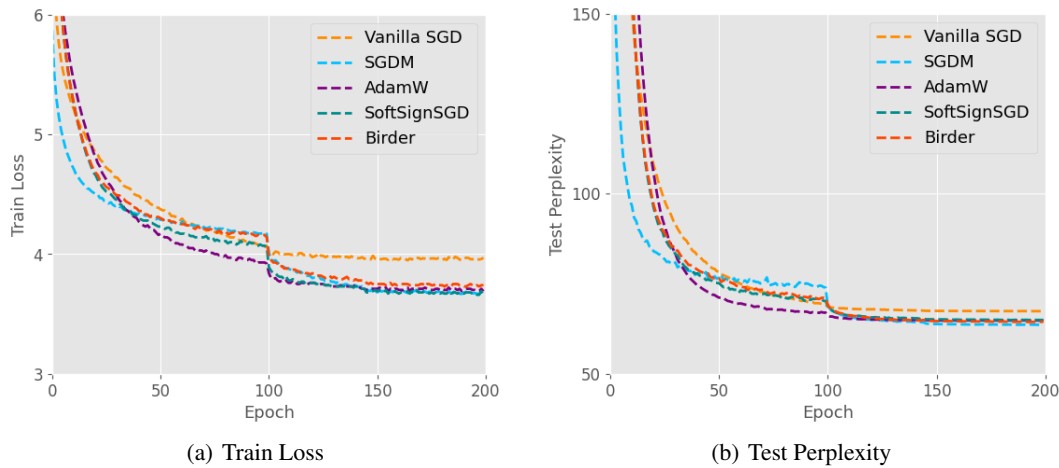

(a) Train Loss

(b) Test Perplexity

Figure 6: Training loss and test perplexity (the lower, the better) for 3-layer LSTM on Penn TreeBank.

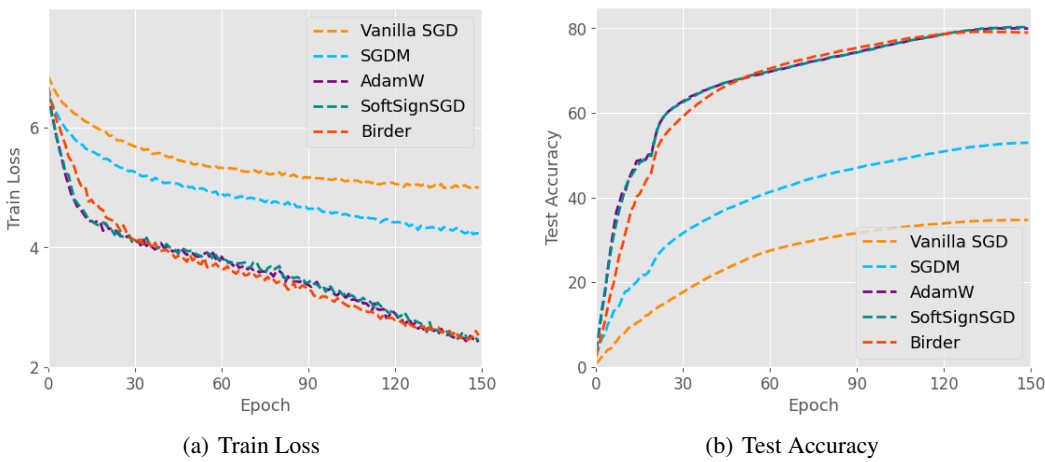

(a) Train Loss

(b) Test Accuracy

Figure 7: Training loss and test accuracy for ViT-B-16 on ILSVRC2012.

## C  Experimental Results for pre-training BERT-Base

We employed BertAdam and Birder to pre-train BERT-Base on Wikipedia using $64$ GPUs ($8$ nodes). The sequence length was set to $512$, and the batch size per GPU was set to $16$. The training process consisted of $37,000$ iterations. The learning rate started at $4 \times 10^{-4}$ and linearly increased in the first $12,500$ iterations, after which it linearly decreased to $0$ for the remaining iterations. For BertAdam, the parameter values $[\beta_1, \beta_2]$ were set to $[0.9, 0.999]$, and for Birder, the momentum parameter beta was set to $0.9$.

As depicted in Figure 8, Birder demonstrates a comparable iteration-wise convergence rate to BertAdam. However, in terms of time-wise convergence, Birder achieves a 4.2x faster convergence rate compared to BertAdam.

## D  Experiments with InfiniBand connections

To further evaluate the communication efficiency of SGDM/Adam, SoftSignSGD and Birder with high bandwidth connections, we implement experiments for training ResNet-50 and BERT-Base with distributed nodes connected with 200Gbps InfiniBand. All the experimental settings are the same as we perform experiments with Ethernet in Subsection 5.1, and the experimental results are listed in Table 3 and Table 4.

As shown in Table 3 and Table 4, compared with the baseline SGDM/Adam, Birder can still reach up to $1.45\times$ speedup for ResNet-50 on ILSVRC2012 and $2.85\times$ speedup for BERT-Base on SQuAD 1.1, although the speed advantage is not so obvious as that with lower-bandwidth Ethernet connections. An interesting phenomenon

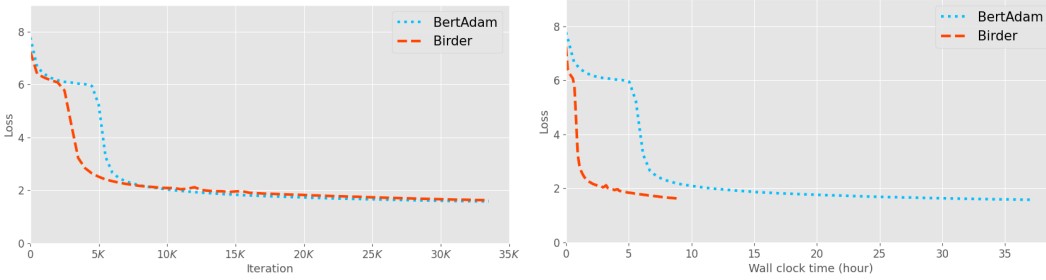

(a) Iteration-wise, BERT-Base, batch size=$16 \times 64$  (b) Time-wise, BERT-Base, batch size=$16 \times 64$

Figure 8: Iteration-wise and time-wise convergence speed for pre-training BERT-Base with 16 samples per GPU with 64 GPUs.

Table 3: System throughput (samples/s) of SGDM, 1-bit Adam and Birder for training ResNet-50 on ILSVRC2012 with 10Gbps Ethernet and 200Gbps InfiniBand.

| #GPUs | Optimizer | Ethernet (10Gbps) | | | InfiniBand (200Gbps) | | |
|---|---|---|---|---|---|---|---|
| | | Throughput (samples/s) | Speedup | Scale Efficiency | Throughput (samples/s) | Speedup | Scale Efficiency |
| 8 | SGDM | 3693 | 1.00× | 100% | 3693 | 1.00× | 100% |
| | 1-bit Adam | 3243 | 0.83× | 100% | 3243 | 0.83× | 100% |
| | Birder | 3462 | 0.94× | 100% | 3462 | 0.94× | 100% |
| 16 | SGDM | 2959 | 1.00× | 40.1% | 4673 | 1.00× | 63.2% |
| | 1-bit Adam | 4715 | 1.60× | 72.7% | 5708 | 1.22× | 88.0% |
| | Birder | 6015 | 2.03× | 86.9% | 6784 | 1.45× | 97.9% |
| 32 | SGDM | 4270 | 1.00× | 28.9% | 9063 | 1.00× | 61.3% |
| | 1-bit Adam | 7268 | 1.70× | 56.0% | 10249 | 1.13× | 79.0% |
| | Birder | 9416 | 2.21× | 68.0% | 12131 | 1.34× | 87.6% |
| 32 | SGDM | 6189 | 1.00× | 20.9% | 16608 | 1.00× | 56.2% |
| | 1-bit Adam | 5546 | 0.89× | 21.3% | 16920 | 1.02× | 65.2% |
| | Birder | 15253 | 2.47× | 55.1% | 19956 | 1.21× | 72.1% |

Table 4: System throughput (samples/s) of BertAdam, 1-bit Adam and Birder for fine tuning BERT-Base on SQuAD 1.1 with 10Gbps Ethernet and 200Gbps InfiniBand.

| #GPUs | Optimizer | Ethernet (10Gbps) | | | InfiniBand (200Gbps) | | |
|---|---|---|---|---|---|---|---|
| | | Throughput (samples/s) | Speedup | Scale Efficiency | Throughput (samples/s) | Speedup | Scale Efficiency |
| 8 | BertAdam | 413 | 1.00× | 100% | 413 | 1.00× | 100% |
| | 1-bit Adam | 358 | 0.87× | 100% | 358 | 0.83× | 100% |
| | Birder | 412 | 1.00× | 100% | 412 | 0.94× | 100% |
| 16 | BertAdam | 84 | 1.00× | 10.1% | 272 | 1.00× | 32.9% |
| | 1-bit Adam | 213 | 2.54× | 29.7% | 522 | 1.92× | 72.9% |
| | Birder | 431 | 5.13× | 52.3% | 776 | 2.85× | 94.1% |
| 32 | BertAdam | 119 | 1.00× | 7.20% | 543 | 1.00× | 32.8% |
| | 1-bit Adam | 274 | 2.30× | 19.1% | 903 | 1.66× | 63.1% |
| | Birder | 730 | 6.13× | 44.2% | 1365 | 2.51× | 82.9% |
| 32 | BertAdam | 158 | 1.00× | 4.78% | 998 | 1.00× | 30.2% |
| | 1-bit Adam | 252 | 1.59× | 8.80% | 1496 | 1.50× | 52.2% |
| | Birder | 990 | 6.26× | 30.0% | 2299 | 2.30× | 69.8% |

is that the system throughput of Birder with 10Gbps Ethernet can match that of SGDM/Adam with 200Gbps InfiniBand.

The experimental results in Table 3 and Table 4 also show that as the number of GPUs is increasing, the scale efficiency of SGDM/Adam, SoftSignSGD and Birder becomes lower. The reason for this phenomenon can be summarized in the following. When the number of GPUs doubles, the number of communication trips also

multiplies. We take the communication scheme *All-Reduce* for example. If the number of GPUs is $n$, each GPU requires $2(n-1)$ trips across the network confections. When the number is non-trivial, the computation time of the communication primitives may exceed the time of the pure communication itself and dominate the overall communication time, since the total communication overhead does not change with the number of GPUs. Notably, *All-reduce* is more efficient than *All-to-All* which is the core of our *Hierarchical-1-bit-All-Reduce*. Hence, as shown in in Table 3 and Table 4, the scale efficiency of Birder decreases more quickly than SGDM/Adam with the number of GPUs growing.

# E    Discussion

The original paper on 1-bit Adam reports a significant speed advantage (up to $3.8\times$) for 1-bit Adam compared to full-precision Adam, with the advantage becoming more prominent as the number of GPUs increases. However, in our experiments, we did not observe clear speed advantages for 1-bit Adam over the original Adam. In fact, when running on 64 GPUs, 1-bit Adam was not only slower than the original Adam, but its throughput rate was even lower than that on 32 GPUs. There are several reasons for this phenomenon. First, the speedup of 1-bit Adam is obtained by comparing the throughput of the compression phase with that of the warm-up phase. However, in our experiments, we evaluated the overall average throughput of both the warm-up phase and the compression phase for 1-bit Adam. Second, the baseline Adam did not run with system-level efficient *DDP*. Third, the authors of 1-bit Adam customized highly efficient communication primitives specifically for their optimizer, whereas we utilized off-the-shelf communication primitives in PyTorch for all the optimizers to ensure fairness.

As shown in Figure 4, as the number of GPUs increases, the communication time for Birder also grows superlinearly. One of the reasons for this is that the communication primitive *All-to-All* accounts for an increasing portion of the communication time. However, the native *All-to-All* in Step (iii) of the *Hierarchical-1-bit-All-Reduce* is not less efficient than the native *All-Reduce*. Therefore, we plan to further optimize the *All-to-All* and *All-Gather* primitives to accelerate Birder.

When training large-scale DNNs, the mixed-precision technique is commonly used to reduce memory consumption, allowing for larger model sizes. While optimizers still utilize full-precision states and computations, which typically contribute to 33-75% of the total memory footprint, Birder does not require full-precision states or computations. Moreover, due to the random quantization of updates to 1 or -1, Birder can leverage lower precision than FP16 gradients to estimate the update. Therefore, Birder shows promise for applications that focus on reducing memory usage, as highlighted in recent research on 8-bit optimizers via block-wise quantization (Tim Dettmers et al., ICLR 2022).

