# OpenReview forum: "Birder: Communication-Efficient 1-bit Adaptive Optimizer for Practical Distributed DNN Training"
_NeurIPS.cc/2023/Conference — NeurIPS 2023 poster_

### Official Review · Reviewer_yyED · 2023-07-03

**Soundness:** 2 fair
**Presentation:** 2 fair
**Contribution:** 2 fair
**Rating:** 5
**Confidence:** 3

**Summary:**

With the increasing size of the model, communication compression plays an important role in distributed machine learning because the model needs to transmit among the nodes. Toward this goal, this paper introduces a one-bit adaptive optimizer named BRAM and further extends the method to hierarchical all-reduce. Specifically, the method is lightly computed because it requires compression only, which significantly saves the time spent on the decompression step. Theoretical analysis shows the efficiency of the proposed algorithm, and the empirical studies validate the method is comparable with one-bit Adam and uncompressed SGDM.

**Strengths:**

**S1.** The authors give a convergence rate of the proposed algorithm, which matches the state-of-the-art approach. Although I do not check the proof details, the result sounds reasonable.

**S2.** The experiment setup is comprehensive. It evaluates ResNet-50 and BERT-based models and conducts experiments on various numbers of GPUs.

**Weaknesses:**

**W1.** The literature review is poor. The authors did not comprehensively survey the works of one-bit compression, e.g., SSDM [1] and Marsit [2]. The unbiased design of Equation (5) has previously been proposed by the method proposed in [1], and [2] discusses a more complex all-reduce case.

**W2.** This work is not very surprising. The existing works on one-bit compression generally use Adam/Momentum as the adapters in their experimental evaluation [1, 2, 3], although their approaches are discussed under SGD settings.

**W3.** As the paper is discussed under the one-bit transmission, the readers expect the proposed method can use single-bit only for each parameter at every transmission. However, the proposed method seems incapable under multihop all-reduce [2] settings.

***

**References:**

[1] Safaryan, Mher, and Peter Richtárik. "Stochastic sign descent methods: New algorithms and better theory." International Conference on Machine Learning. PMLR, 2021.

[2] Wu, Feijie, et al. "Sign bit is enough: a learning synchronization framework for multi-hop all-reduce with ultimate compression." Proceedings of the 59th ACM/IEEE Design Automation Conference. 2022.

[3] Bernstein, Jeremy, et al. "signSGD with Majority Vote is Communication Efficient and Fault Tolerant." International Conference on Learning Representations. 2018.

**Questions:**

**Q1.** The introduction claims that the proposed algorithm performs similarly to uncompressed SGDM/Adam. However, the experiments compare the proposed work with one-bit Adam only. Is it over-claimed because we cannot see how close it is between the proposed work and uncompressed Adam? Can the authors provide the results for uncompressed Adam?

**Limitations:**

**L1.** The method designed for Hierarchical-1-bit All-Reduce cannot guarantee every transmission using 1 bit only. It will require extra steps to encode the transmitted gradients at the sender and decode them at the receiver.

---

> ### Author Rebuttal · Authors · 2023-08-09
>
> We express our sincere gratitude for your volunteering time and your valuable comments. In response to your feedback, we have organized our response into the following points:
>
> **Q1.** *The literature review is poor. The authors did not comprehensively survey the works of one-bit compression, e.g., SSDM [1] and Marsit [2]. *
>
> **R1.** Thank you for bringing Ref [1-2]. Due to space limitations, our manuscript mainly discussed classic methods like signSGD,  TernGrad, QSGD, DIANA, 1-bit Adam, etc.,  did not cover variants of signSGD such as SSDM[1] and Marsit[2]. We guess you might misunderstand our proposed BRAM was also a variant of signSGD, so we shall  discuss SSDM and Marsit. As we emphasized in Section 2 in detail, BRAM is not a variant of signSGD. Thus, it might not be necessary to survey  [1-2].
>
>
> **Q2:** *The existing works on one-bit compression generally use Adam/momentum as the adapters in their experimental evaluation [1-3].*
>
> **R2:**  We respectfully disagree that all existing 1-bit compression optimizers are closely resemble those discussed in [1-3]. While character limitation  prevents an exhaustive comparison between the methods in [1-3] and prior classic methods, we focus on  distinctions between the methods in [1-3] and BRAM. The methods in [2-3] are essentially signSGD,  and we have extensively examined the differences between signSGD and BRAM in Section 2. Marsit, as presented in [2], introduces a 1-bit data communication scheme, and the contrast between Marsit and our Hierarchical-1-bit-All-Reduce is elaborated in **R3**.  Signum (signSGD witht momentum)  in [3] markedly  differ from that of BRAM. This dissimilarity  can be also validated by the experimental results; [3] shows that Signum  exhibits significantly inferior performance to Adam, while our paper demonstrate BRAM achieves comparable and even superior performance, compared  to Adam.
>
> Below, we will focus on a comprehensive analysis of the distinctions between BRAM and SSDM [1], addressing your foremost concerns.
>
> BRAM can be simplified as
> $$
> m_{t} = \beta m_{t-1} + (1-\beta)g_t,
> b_{t}  = \beta b_{t-1} + (1-\beta)|g_t|,
> x_{t+1}  = x_t - \alpha_t {\mathcal Q_B} \left(\frac{m_{t}}{b_t}\right),
> $$
> where ${\mathcal Q _B}(\cdot)$ is element-wise , and it quantizes the $j$-th element  as
> $$
> {\mathcal Q_B} \left(\frac{(m _{t})_j}{(b _t)_j} \right) =
> \begin{cases}
> 1& {\rm with ~ prob}~ p= \frac{1}{2}(\frac{(m _{t})_j}{(b _t)_j}+1) \\\\
> -1& {\rm with ~ prob}~ 1-p
> \end{cases}
> $$
>
> SSDM can be simplified as
> $$
> m_{t}  = \beta m_{t-1} + (1-\beta)g_t, \\\\
> x_{t+1} = x_t - \alpha_t {\mathcal Q_S} \left(\frac{m_{t}}{\Vert m_t \Vert_2}\right),
> $$
>
> where ${\mathcal Q_S}(\cdot)$ is  vector-wise,  *i.e.*, :
> $$
> {\mathcal Q_S}\left(\frac{(m _{t})_j}{\Vert m_t \Vert_2}\right) =
> \begin{cases}
> 1&  {\rm with ~ prob}~ p= \frac{1}{2}(\frac{(m _{t}) _j}{\Vert m_t \Vert_2}+1)\\\\
> 0& (m_t)_j=0\\\\
> -1& {\rm with ~ prob}~ 1-p
> \end{cases}
> $$
>
> The primary divergence lies in their quantization, where ${\mathcal Q_B}(\cdot)$ in BRAM is element-wise, while ${\mathcal Q_S}(\cdot)$ in SSDM is vector-wise.  As for BRAM,   ${\mathbb E}\left(\alpha_t{\mathcal Q_B}\left(\frac{(m _{t})_j}{(b _t)_j}\right)\right) =  \frac{\alpha_t}{(b _t)_j}\cdot(m _{t})_j$ where $\frac{\alpha_t}{(b _t)_j}$ is the adaptive learning rate, just like Adam. In contrary, SSDM maintains a uniform learning rate $\frac{\alpha_t}{\Vert m_t \Vert_2}$ due to its vector-wise quantization.  Moreover,  the dimension of $m_t$ for DNNs is much large, so that $\frac{(m _{t})_j}{\Vert m_t \Vert_2} \approx 0$, which will cause $x_t$ to mostly stay constant, rendering it unsuitable for training large DNNs. Conversely, BRAM's quantization ensures that each update is independent of the dimension of $m_t$. In brief, the contrast empowers BRAM to achieve quicker convergence and superior performance compared to SSDM.
>
>
> Another differentiation is that ${\mathcal Q_B}(\cdot)$  employs 2 states ($-1$ and $1$), while ${\mathcal Q_S}(\cdot)$ employs 3 states ($-1$, $0$, and $1$). This means that BRAM requires fewer communication bits compared to SSDM.  Consequently, BRAM is more communication-efficiency than SSDM.
>
> More importantly, the vector-wise quantization in SSDM can trace back to earlier work QSGD. In contrast, BRAM is the first adaptive optimizer that element-wise quantizes the entire update.
>
>
> **Q3.** *The method designed for Hierarchical-1-bit All-Reduce cannot guarantee every transmission using 1 bit only. The proposed method seems incapable under multihop all-reduce [2] settings.*
>
> **R3.** We respectfully disagree again with you on this issue. You might have skipped Section 4 and Figure 1.  They depict in detail how to communication 1-bit data with multi-hop communication scheme via Hierarchical-1-bit-All-Reduce.
>
> Moreover, let's consider Marsit's methodology. Marsit can only guarantee the expected value of the final 1-bit data equals to the average of  the original 1-bit data among nodes, thereby it will bring performance deterioration. In contrast, Hierarchical-1-bit-All-Reduce ensures the final  data exactly equals to  the average of  the original data. Hierarchical-1-bit-All-Reduce effectively utilizes primitives natively supported by NCCL and Gloo, while Marsit needs to customization. Thus, Hierarchical-1-bit-All-Reduce is also more efficient than Marsit.
>
> Notably, the latest influential work ZeRO++  incorporated a method taht is nearly identical toHierarchical-1-bit-All-Reduce for LLM training . This demonstrates its effectiveness and generalization for large-scale distributed training.
>
> **Q4.** *There are no experimental results for uncompressed Adam.*
>
> **R4.** We appreciate your attention to our experiments. We want to highlight that our experimental results for uncompressed Adam are comprehensively presented in Section 5.2, Table 2, and Figure 2(e-f) of the main text. Additionally, more extensive results for  it are provided in Section B and Figures 1-4 in the Appendix.

---

> > ### Author Response · Authors · 2023-08-16
> > **Looking Forward to Further Discussion**
> >
> > Dear Reviewer,
> >
> > We highly appreciate your valuable comments. In our reponse,  we hope that your concerns have been addressed. We would like to discuss with you in this period and are willing to provide more according to your feedback or further questions.
> >
> > If you feel happy with our response, please consider updating your score. Feel free to reach us out in case you need any clarification.
> >
> > Thanks,
> >
> > Paper 5560 Authors

---

> > ### Comment · Reviewer_yyED · 2023-08-18
> > **Response**
> >
> > Thanks a lot for your response, and I apologize for my late reply. After reading your response, I have a few more comments:
> >
> > **P1.** The papers I share with you are **very** related to your works. Quantization and signSGD are the most popular methods to achieve one-bit transmission, but they are not independent. Strictly speaking, SSDM is not the variant of signSGD because it does not simply use the sign of an element. Besides, Definition 3 of SSDM follows the idea where an element $x \in [-1, 1]$ represents with +1 with the probability of $\frac{1}{2}(1+x)$ or -1 otherwise, which is very similar to your Equation (5). Last but not least, I agree with your explanation of why SSDM does not perform well in DNN, and I hope you can include the comparison in your revision.
> >
> > **P2.** What I mean in **W3** is that the proposed algorithm cannot achieve one-bit communication for intra-node (i.e., the Reduce-scatter step of Figure 1). Also, an "all-to-all" communication pattern (a.k.a., complete graph) is likely to cause network conjunction because a node should handle multisource information at once [1]. According to the definition of multihop all-reduce [2], it seems that the proposed algorithm is not a good solution to the scenarios like ring all-reduce.
> >
> > As you mention LLM, I have another concern about your work. Suppose each node is capable of training a single LLM. Since "all-to-all" is used in Hierarchical-1-bit-All-Reduce, some packets are inevitably lost. How can you effectively handle the network issue? Will the proposed algorithm require the sources to retransmit the packets?
> >
> > **P3.** I have a new question regarding your Theorem 1. I am not sure whether it is a typo that the LHS of Equation 6 should be $\mathbb{E} [\frac{1}{T} \sum_{t=1}^T ||\nabla f(x_t)||^2]$. If not, I don't think you can make the remark between Line 177 --178 because $\mathbb{E} [\frac{1}{T} \sum_{t=1}^T ||\nabla f(x_t)||^2]$ is greater than the LHS of Equation 6, according to Cauchy–Schwarz inequality.
> >
> > **References:**
> >
> > [1] Sergeev, Alexander, and Mike Del Balso. "Horovod: fast and easy distributed deep learning in TensorFlow." arXiv preprint arXiv:1802.05799 (2018).
> >
> > [2] Wu, Feijie, et al. "Sign bit is enough: a learning synchronization framework for multi-hop all-reduce with ultimate compression." Proceedings of the 59th ACM/IEEE Design Automation Conference. 2022.

---

> > > ### Author Response · Authors · 2023-08-18
> > > **Responses to the Post-Rebuttal Feedback (1/2)**
> > >
> > > We sincerely appreciate your valuable feedback. To address your additional concerns, we have organized our response into the following key points:
> > >
> > > **Q5.**  *The comaprison between the quantion mehtods of SSDM and BRAM.*
> > >
> > > **R5.**  Sincerely Thank you for your continued recommendation of references, and we will dicussed them in the upcoming revision. It's important to clarify that our paper does not claim the quantization technique involving elements $x \in [-1, 1]$ being quantized to $1$ with a probability of $p=\frac{x+1}{2}$ (or $-1$ otherwise) as a novelty, since we are aware that this technique is straightforward and can trace back to earlier classical works like QSGD and TernGrad.
> > >
> > >  In contrast,  as highlighted consistently in both our paper and prior responses, our primary contribution in BRAM's quantization lies in being the first to **element-wise** quantize the entire update of an adaptive optimizer.  In comparison to existing communication-quantization optimizers (including SSDM) that conduct **vector-wise** quantization for updates, BRAM offers not only the benefit of **adaptive preconditioning** for rapid training speed and good inference performance, just like  Adam, but also enhances distributed communication efficiency through light element-wise quantization.
> > >
> > > In summary, *we still believe that there are significant divergences between the quatization approches of SSDM and BRAM*. As you suggested, we will comprehensively analyze and compare SSDM and BRAM in the upcoming revision.
> > >
> > > **Q6.** *Concerns regarding Herarchical-1-bit-All-Reduce.*
> > >
> > > **R6.** Thank you for the further comments about Herarchical-1-bit-All-Reduce. We respectfully disagree with the notion that Herarchical-1-bit-All-Reduce cannot achieve one-bit communication among intra nodes in the initial stage. In fact, we can employ intra-node 1-bit All-to-All along with local averaging and requantization to **realize 1-bit communication**—a methodology simlar to ZeRO++'s [1] . Substituting Reduce-Scatter with 1-bit All-to-All + local averaging & requantization could potentially result in precision loss due to the introduction of additional requantization steps. As we emphasized  in both Section 1 and Section 3, our selection of Reduce-Scatter serves to maximize the exploration of ultra-high intra-node bandwidth, all while ensuring that precision remains intact and unaffected by the process.
> > >
> > > We would like to further clarify that, as shown  in Algorithm 1 below and in the figure  at the anonymous link (https://1drv.ms/i/s!Au3MrR-o69M4h_oAuUy0GAcIVEWvxg?e=GPMwGm), **All-to-All involves each node sending data to one node while simultaneously receiving data from one node,  just like Reduce-Scatter**. This characteristic ensures that All-to-All doesn't concurrently handle multisource information, thus mitigating any potential for network congestion—addressing your concern. Notably, ZeRO++ [1] has effectively employed All-to-All to communicate quantized data during large-scale distributed LLM training, demonstrating the practical validity of the approach.
> > >
> > > Additionally, just like all the other communication primitives, such as All-Reduce and All-Gather,  All-to-All is built on TCP/IP or RDMA Protocal.  We believe that the TCP/IP or RDMA Protocol can facilitate packet retransmission if any packets are lost during communication, even though the specifics of lower-layer protocols are not within our realm of familiarity.  It's worth noting that these higher-layer communication primitives remain unaware of these lower-layer intricacies.
> > >
> > > | Algorithm 1. Implementation of All-to-All|
> > > |------------|
> > > **Input:** the comunication data  $InputD$, the rank  of the node $r$,  and the number of the nodes $m$
> > > Slice $InputD$ to $n$ segments as $D[0], D[1], ..., D[n]$
> > >
> > >  **for** $i=1.. n-1$ **then**
> > >
> > >   &emsp;  $j \leftarrow (r + i ) ~{\rm mod} ~n$
> > >
> > >   &emsp; $SendD$ $\leftarrow$ send $D[j]$ to the node with the rank $j$
> > >
> > >  &emsp; $k \leftarrow (r + n - i ) ~{\rm mod} ~n$
> > >
> > >  &emsp;  $RecvD$ $\leftarrow$ receive data from the node with the rank $k$
> > >
> > >  &emsp;  $D[k] \leftarrow RecvD$
> > >
> > >  **end for**
> > >
> > >  Concatenate $D[0], D[1], ..., D[n]$  to $OutputD$
> > >
> > > |**Output** $OutputD$ &emsp; &emsp;&emsp;&emsp;&emsp;&emsp;&emsp;&emsp;&emsp;&emsp;&emsp;&emsp;&emsp;&emsp;&emsp;&emsp;&emsp;&emsp;&emsp;&emsp;&emsp;&emsp;&emsp;&emsp;&emsp;&emsp;&emsp;&emsp;&emsp;&emsp;&emsp;&emsp;&emsp;&emsp;&emsp;|
> > > |-----|
> > > ||
> > >
> > >
> > > [1] G. Wang, et al. "ZeRO++: Extremely Efficient Collective Communication for Giant Model Training." arXiv:2306.10209, 2023.

---

> > > > ### Author Response · Authors · 2023-08-18
> > > > **Responses to the Post-Rebuttal Feedback (2/2)**
> > > >
> > > > **Q7.** *A new question regarding the LHS of Equantion (6).*
> > > >
> > > > **R7.**  Thank you for this new comment. In reference to the left-hand side (LHS) of Equation (6), we kindly direct your attention to Appendix, Lines 66-72 (Theorem 1's proof).  From that, we can easily obtaint that ${\mathbb E}\left[\frac{1}{T}\sum_{t=1}^T \Vert \nabla f(x_t) \Vert\right]^2 \le \frac{1}{T}\sum_{t=1}^T {\mathbb E}\left[ \Vert \nabla f(x_t) \Vert^2\right] = O(\frac{1}{\sqrt{T}})$ just due to Cauchy-Schwarz inequality. **This enables us to confidently assert that BRAM attains the same theoretical convergence rate as Adam —addressing your concern.**
> > > >
> > > > Notably, in the optimization community,  using ${\mathbb E}\left[\frac{1}{T}\sum_{t=1}^T \Vert \nabla f(x_t) \Vert\right]$ or  $\frac{1}{T}\sum_{t=1}^T {\mathbb E}\left[ \Vert \nabla f(x_t) \Vert^2\right] $on the LHS to showcase the convergence rate of a non-convex optimization problem is **based on authors' preferences**. Importantly, this choice does not impact the actual convergence rate. In fact, while the authors of  Adam may prefer to $\frac{1}{T}\sum_{t=1}^T {\mathbb E}\left[ \Vert \nabla f(x_t) \Vert^2\right] $ ,   other works like [1-3]  opt for  the formulation of ${\mathbb E}\left[\frac{1}{T}\sum_{t=1}^T \Vert \nabla f(x_t) \Vert\right]$ .
> > > >
> > > > [1] C. Fang, et al. "Near-optimal non-convex optimization via stochastic path-integrated differential estimator." NeurIPS, 2018.
> > > >
> > > > [2] C. Ashok and F. Orabona. "Momentum-based variance reduction in non-convex sgd."  NeurIPS, 2019.
> > > >
> > > > [3] F. Huang, et al. "Super-adam: faster and universal framework of adaptive gradients."  ICML, 2021.

---

> > > > > ### Comment · Reviewer_yyED · 2023-08-20
> > > > > **Thanks for your response**
> > > > >
> > > > > Thanks a lot for the author's response. My concerns have been well addressed. However, I cannot see a **very clear novelty** from the algorithm design (i.e., Adam and error compensation have been proposed in the previous works) and the theorem (e.g., a significant improvement in terms of the convergence rate). And the authors miss some very relevant citations in the first draft. Although the proposed algorithm performs well in the empirical studies, I think this work is a borderline one.
> > > > >
> > > > > There is a reason why I eventually decide to raise my rating from 4 to 5: I find it interesting to use Adam-like update to achieve one-bit communication, together with error compensation, which reduces the sparsity among different elements' value and makes the one-bit compression more effective. However, as I mention above, this work is borderline, and I am less certain if this work deserves acceptance.

---

> > > > > > ### Author Response · Authors · 2023-08-21
> > > > > > **Thank You for Rasing Rating and Some Further Clarifications**
> > > > > >
> > > > > > Many Thanks for raising the score; we're pleased that your concerns have been effectively addressed.
> > > > > >
> > > > > > Regarding the novelty of this paper, we would like to provide further clarifications for your consideration :
> > > > > >
> > > > > > - While existing adaptive optimizers of the Adam type employ a square root of variance-like momentum for updates, utilizing distinct exponential moving average factors for the numerator and denominator, our BRAM differentiates itself.  It employs element-wise absolute-like momentum as the denominator, sharing a uniform exponential moving average factor for both numerator and denominator. These designs reduce computational costs and confine updates to the range of $[-1, 1]$, benefiting light quantization and efficient communication. Additionally, they significantly minimize the required tuning effort. In summary, **compared to existing Adam variants, BRAM offers deeper insights into the designs of adaptive optimizers, particularly  those communcaiton-compression optimzres.**
> > > > > >
> > > > > > -  The previous work [1]  has theoretically  proven  that under $L$-Lipschitz gradident assumption (as discripted in Assumption 2 in our paper),  any  non-convex first-order stochastic optimzation algorithm requires $\epsilon^{-4}$ steps to find a $\epsilon$-stationary point (with the gradient norm at most $\epsilon$, *i.e*, ${\mathbb E}[\Vert \nabla f(x) \Vert] \le \epsilon$), which means the convergence rate for these algorithms is at most  ${\mathbb E}\left[\Vert \nabla f(x_t) \Vert\right]^2 = O( \frac{1}{\sqrt{T}}) $ .  Considering  Adam has already achieved  the covnergence rate $ O( \frac{1}{\sqrt{T}}) $, it might be impossible for BRAM to improve the covergence rate. In fact, all the other adaptive optimizers, such as AMSGrad,  AdamW, AdaBelief, AdaBound, LAMB, Adan and so on,  are also proven to only attain this covergence rate  of $ O( \frac{1}{\sqrt{T}}) $. Notably,  **BRAM's distinct significance lies in achieving the same convergence rate as uncompressed Adam, even when employing extreme 1-bit quantization.**
> > > > > >
> > > > > > [1] Y. Arjevani, el al. "Lower bounds for non-convex stochastic optimization". Mathematical Programming, 2022.

---

### Official Review · Reviewer_jDqW · 2023-07-06

**Soundness:** 3 good
**Presentation:** 3 good
**Contribution:** 3 good
**Rating:** 5
**Confidence:** 3

**Summary:**

The authors propose a novel 1-bit adaptive optimizer by compressing the total update in each iteration. Besides, the author modifies the adam update with the absolute value of the gradient. No bias correction is needed and error feedback is applied in algorithm 1. Hierarchical-1-bit-All-Reduce is proposed, where intra-node communication is full precision, while inter-node communication is compressed. Experiments verifies that the proposed method achieves faster convergence and the model performance is roughly the same.

**Strengths:**

- experiments show wall-clock time acceleration under different local batch size settings. Typically, when local batch size is larger, the computation to communication ratio is higher, and communication compression will lead to lower speedup compared with full-precision training.
- Hierarchical-1-bit-All-Reduce is proposed and sounds reasonable, since the intra-node communication could have a higher bandwidth.
- the wall-clock training time comparison explicitly show the faster convergence speed of the proposed method.

**Weaknesses:**

- all-gather instead of all-reduced is used in figure 1
- noticeable performance drop could be observed from Table 1
- gradient norm needs to be bounded in assumption 4
- the theoretical analysis is rather short and lacks insight.

**Questions:**

- m and b are not synchronized across workers. how do you make sure that the workers do not diverge because of different b?
- have you tried signsgd instead of Eq(5)? Error feedback could work better for biased compressor.
- how do you get rid of batch size in convergence results, which is considered as a theoretical difficulty in Adam?

**Limitations:**

The authors did no discuss societal impact. I don't see potential negative societal impact neither.

---

> ### Author Rebuttal · Authors · 2023-08-09
>
> We express our sincere gratitude for your volunteering time and your valuable comments. In response to your feedback, we have organized our response into the following points:
>
> **Q1.**  *All-gather instead of All-Reduce is used in figure 1*
>
> **R1.** Thank you for raising this concern.  We hold a respectful difference of opinion that Hierarchical-1-bit-All-Reduce just simply substitutes All-Reduce with All-gather.  In fact, two key innovations define Hierarchical-1-bit-All-Reduce. First, it hierarchically harnesses the ultra-fast intra-node bandwidth to boost local communication efficiency. Second, it originally introduces 1-bit-All-Reduce to further reduce communication overhead across inter-nodes.
>
> While the traditional All-Reduce comprises Reduce-Scatter and All-Gather.  We recognize the limitations of Reduce-Scatter in the context of 1-bit communication due to potential overflow.  Consequently, as elaborated in Section 3 and visually displayed in Figure 1, our work has ingeniously proposed the concept of 1-bit-All-Reduce. This innovative approach substitutes Reduce-Scatter with a combination of 1-bit All-to-All , local average and re-quantization and All-Gather. Assuming we have $n$ nodes, and  the overall volume for each GPU needs to be communicated is $P$, the communication complexity of All-to-All and All-Gather in 1-bit-All-Reduce is $\frac{(n-1)P}{n}$, which equals to that of the ideal 1-bit ring-based All-Reduce without overflow. In contrast, the complexity of simply subsituting All-Redeuc with All-Gather is $(n-1)P$.
>
> **Q2.** *Noticeable performance drop could be observed from Table 1*
>
> **R2.** We appreciate your attention to detail. We followed the PyTorch official source codes  for training ResNet-50 using SGDM and BRAM. Noting that a decline in test accuracy with larger batch sizes is a well-established phenomenon, as demonstrated in prior research [1]. Our focus is on enhancing training efficiency, rather than chasing state-of-the-art performance with large batch sizes.
>
> **Q3.** *Gradient norm needs to be bounded in Assumption 4*
>
> **R3.**   Assumption 4 is commonly used to analyze the theoretical aspects of adaptive optimizers, *e.g.*, Adam and its variants [2-5]. Notably, Assumption 4 is typically met during the training of DNNs in practical scenarios.
>
> **Q4.** *The theoretical analysis is rather short and lacks insight*
>
> **R4.**  Our theoretical convergence analysis for BRAM spans approximately 9 pages, which we consider to be a substantial coverage. Notably, the length of the theoretical analysis may not necessarily correlate with its theoretical significance and depth of insight.
>
> **Q5.** *$m_t^{(i)}$ and $b_t^{(i)}$ are not synchronized across workers. How do you make sure that the workers do not diverge?*
>
> **R5.** We appreciate your attention to this matter. Despite the potential lack of synchronization for $m_t^{(i)}$ and $b_t^{(i)}$ among workers, it's important to note that the parameter $x_t$ achieves synchronization across all workers with each iteration. Furthermore, the independently and i.i.d. nature of the sampled data on each worker guarantees that $m_t^{(i)}$ and $b_t^{(i)}$ are inherently i.i.d., thus ensuring theoretical convergence. Notably,  Signum with majority vote (the momentum version of signSGD) [6] also doesn't necessitate synchronization for momentum across workers, and it theoretically proves that Signum with majority vote will be converged.
>
> **Q6.** *Have you tried signSGD instead of Eq(5)? Error feedback could work better for biased compressor.*
>
> **R6.** We did not explore the utilization of signSGD with error feedback (EF-signSGD) [7] as a replacement for Eq. (5). The main reason is that EF-signSGD, like signSGD,should  employs a majority vote mechanism to aggregate data across workers,  rendering it compatible solely with the less efficient All-Gather instead of All-Reduce.  Consequently, its applicability for practical large-scale distributed training  becomes limited, which has been validated in previous studies [8]. Moreover, EF-signSGD aligns more closely with SGD-type optimizers rather than Adam-type optimizers, making it inapplicable to train Transformer-based DNNs, as we discussed in Section 1. This is also experimentally validated in figure 1 presented in the "globe response" PDF file.
>
> **Q7.** *How do you get rid of batch size in convergence results, which is considered as a theoretical difficulty in Adam?*
>
> **R7.**  Thank you for taking our attention to this crucial problem that we were not aware of. Actually, we did not fully get rid of the mini-batch size $m$ in the convergence analysis. In Assumption 4, we assume   $\Vert g_t^{(i)} \Vert \le G$. It is known that ${\mathbb E} [\Vert g_t^{(i)} \Vert^2] = \Vert {\mathbb E} [g_t^{(i)} ]\Vert ^2 + {\mathbb E} [\Vert g_t^{(i)} -  {\mathbb E} [g_t^{(i)} ] \Vert^2] = \Vert {\mathbb E} [g_t^{(i)} ]\Vert ^2 + \Vert {\mathbb V} [g_t^{(i)} ]\Vert ^2$. $\Vert {\mathbb E} [g_t^{(i)} ]\Vert ^2$ is commonly small, and $\Vert {\mathbb V} [g_t^{(i)} ]\Vert ^2$ is inverse proportion to the mini-batch size $m$. Thus,  $G$ is inherently interconnected with the mini-batch size $m$.
>
>
> [1] Y. You, et al. "Large batch training of convolutional networks." arXiv:1708.03888, 2017.
>
> [2] Diederik P. Kingma, and. Ba. "Adam: A method for stochastic optimization."  arXiv:1412.6980, 2014.
>
> [3] M.  Zaheer et al. "Adaptive methods for nonconvex optimization." NeurIPS, 2018.
>
> [4] L. Luo , et al. "Adaptive gradient methods with dynamic bound of learning rate. ICLR, 2019.
>
> [5] Y. You, et al. "Large batch optimization for deep learning: Training bert in 76 minutes." ICLR, 2020.
>
> [6] J. Bernstein, et al. "signSGD with majority vote is communication efficient and fault tolerant." ICLR, 2019.
>
> [7] SP Karimireddy, et al. "Error feedback fixes signsgd and other gradient compression schemes." ICML, 2019.
>
> [8] S. Agarwal , et al. "On the utility of gradient compression in distributed training systems." MLSys, 2022.

---

> > ### Author Response · Authors · 2023-08-16
> > **Looking Forward to Further Discussion**
> >
> > Dear Reviewer,
> >
> > We highly appreciate your valuable comments. In our reponse,  we hope that your concerns have been addressed. We would like to discuss with you in this period and are willing to provide more according to your feedback or further questions.
> >
> > If you feel happy with our response, please consider updating your score. Feel free to reach us out in case you need any clarification.
> >
> > Thanks,
> >
> > Paper 5560 Authors

---

> > > ### Author Response · Authors · 2023-08-21
> > > **Eagerly Expecting a Last Minute Discussion**
> > >
> > > As the author-reviewer discussion deadline inchingly near, we are looking forward to engaging in a quick discussion with you. Please don't hesitate to reach us if you require any clarifications or further assistance.
> > >
> > > If our responses have addressed your concerns, we would greatly appreciate it if you could consider updating your rating.
> > >
> > > Thanks,
> > >
> > > Paper 5560 Authors

---

### Official Review · Reviewer_5r9C · 2023-07-11

**Soundness:** 3 good
**Presentation:** 3 good
**Contribution:** 2 fair
**Rating:** 4
**Confidence:** 4

**Summary:**

This paper proposes a compute-efficient gradient compressor named BRAM which randomly quantifies the update direction into 1-bit for communication. A theoretical convergence bound is also provided to prove BRAM’s convergence property. To improve communication efficiency, BRAM also integrates a novel Hierarchical-1-bit-All-Reduce algorithm that can better utilize high-bandwidth intra-node communication. Experiments conducted on two popular DNNs (ResNet-50 with ImageNet and BERT-Base with SQuAD1.1) show BRAM seems to outperform SGDm, Adam, and 1-bit Adam.

**Strengths:**

1. A compute-efficient lightweight compressor on model updates (not gradients) is proposed to reduce communication volume in distributed data-parallel training.
2. A novel *Hierarchical-1-bit-All-Reduce* algorithm is proposed to better utilize the high-bandwidth intra-node communication and alleviate the communication through low-bandwidth inter-node interconnect.

**Weaknesses:**

1. Experiments are not strong to support the claims of the paper.
2. Some state-of-the-art compressed methods like HiTopKComm (with MSTop-k) and PowerSGD were not compared.

**Questions:**

1. Experiments seem not to be conducted fairly. 1) The ResNet-50 model was not trained with standard accuracy (i.e., 75.9% top-1 accuracy according to MLPerf and many other benchmarks). For example, in Table 1, top-1 accuracy drops to 74.+% on 64 GPUs with 128 samples per GPU (i.e., a global batch size of 8K), but many studies show ResNet-50 on ImageNet can be scaled to a mini-batch size of 32K to reserve 75.9% top-1 accuracy. 2) The testbed has a very high intra-node bandwidth but relatively low inter-node bandwidth, thus the *Hierarchical-1-bit-All-Reduce* algorithm is important. However, the experimental setting seems deliberately constructed to show the effectiveness of BRAM according to Table 1 where only 128 samples were used for each Nvidia A100 GPU such that the computation-to-communication ratio is low. To fairly compare with SGDM, the local batch size is possible to be 512 or 1024 on an Nvidia A100-80G GPU. It is similar to the BERT-Base experiments, where the local mini-batch size is only 3. Is it out of memory if setting a larger mini-batch size?

2. The paper claims that BRAM doesn’t need a warmup strategy to align the accuracy to the dense version, but the benchmarks don’t have standard accuracy. Is it possible that 1-bit Adam, MSTop-k (with HiTopKComm), and PowerSGD also achieve similar accuracy achieved by BRAM even if they train DNNs without warmup? How about comparing MSTop-k and PowerSGD in terms of end-to-end training time to achieve target accuracy?

3. It seems that the convergence analysis in Section 3 is based on the theoretical framework from [8]. Is the theoretical convergence analysis providing new insights in this paper compared with [8]?

4. The *1-bit-All-Reduce* algorithm converts all-reduce to all-to-all, how about the communication complexity of *Hierarchical-1-bit-All-Reduce* compared to state-of-the-art all-reduce algorithms (e.g., ring-based, tree-based, etc.)? A communication complexity comparison is suggested to be provided to help understand the advantage of  Hierarchical-1-bit-All-Reduce.

5. The key idea of the *Hierarchical-1-bit-All-Reduce algorithm* is almost the same with HiTopKComm in [23] though they are used under different scenarios. Should credit be given to [23]? and what are their main differences?

**Limitations:**

See above.

---

> ### Author Rebuttal · Authors · 2023-08-09
>
> We express our sincere gratitude for your volunteering time and your valuable comments.  We organize our response into the following points:
>
> **Q1.**  *The top-1 accuracy drops to 74.+\% with globe batch size of 8k for training ResNet-50, while methods in MLPerf can reserve 75.9\% with batch size of 32k.*
>
> **R1.**  In response to your observation, we carefully examined the source codes of the top-20 methods on MLPerf. Our investigation revealed  that all these methods select the specifically designed LARS for large batch-size training.  In fact, a decline in test accuracy with larger batch sizes is a well-established phenomenon, as demonstrated in the paper on LARS [1].  Thank you for your reminder,   we will incorporate the LARS technique into BRAM in our upcoming research.
>
> **Q2.** *128 samples per GPU were used to train ResNet-50, and 3 samples per GPU were used to finetune BERT.*
>
> **R2.** Thank you for sharing your acute observation.  The decision regarding the number of samples per GPU  followed closely-related prior papers and  source code tutorials. [2-4] train ResNet on IMAGENET with 64, 128 and 256 samples per GPU respectively to show the efficiency of communicaiton-compression optimizers, and the HuggingFace tutorial (https://huggingface.co/bert-base-uncased) and the paper [3] pretrain BERT on  English Wikipedia with 16 samples per GPU and finetune BERT on SQuAD with 3 samples per GPU. In alignment with these established practices, we  chose  a sample count of 128 per GPU for training ResNet-50, and 16/3 samples for the pretraining/finetuning BERT.
>
> **Q3.** *Comparison to MSTop-k and PowerSGD in terms of end-to-end training time.*
>
> **R3.**   It seems there might be some misunderstandings. In the section of Introduction, our intention was to emphasize that some rather than all existing communication-compression optimizers require warmup.
>
>  Both in the Section 1 and Section 5, we've highlighted the findings from recent works [2][5]  that  PowerSGD and MSTop-k  are an overall slower than SGDM/Adam,  when employed with  DDP.  Specifically, PowerSGD incurs heavy compression/decompression computation costs that outweigh the communication reduction gains, while MSTop-k, when coupled with HiTopKComm, encounters incompatibility with All-Reduce and resorts to AllGather, which significantly escalates the communication complexity.  Therefore, considering that our paper focuses on communication efficiency and we have compared BRAM with uncompressed SGDM/Adam, we thought a direct comparison with PowerSGD and MSTop-k might not be necessary.
>
> **Q4.** Is the theoretical convergence analysis providing new insights compared with Ref (8)?
>
> **R4.** We agree with you that the convergence analysis is inspired by Ref (8).However,  the theoretical analysis for BRAM encompasses more intricate elements, including  local/global quantization  and error feedback, along with the fixed and uniform $\beta$ for $m_t$ and $b_t$. Consequently, these factors render the theoretical investigation  more complicated than that in Ref (8).  Notably,  the theorectical convergence analyses of numerous recent works, such as ZO-AdaMM [6], AdaBelief [7], and FedAMS[8], were also motivated by Ref (8). However, this does not affect the theoretical significance or contributions of these individual studies.
>
> **Q5.** *The main differences and the communication complexity comparison between Hierarchical-1-bit-All-Reduce and HiTopKComm.*
>
> **R5.** The notion of hierarchicy in Hierarchical-1-bit-All-Reduce are from HiTopKComm, but we respectfully disagree with you that  Hierarchical-1-bit-All-Reduce is almost the same with HiTopKComm.  We originally propose 1-bit-All-Reduce to communication data cross inter-nodes with the bottleneck bandwidth.  In contrast, HiTopKComm relies on existing All-Gather in a more straightforward manner. Specifically,  1-bit-All-Reduce combines sequential processes, including All-to-All, local averaging and re-quantization, and All-Gather. As detailedly analyzed in Section 3, when considering $n$ nodes and an communication volume of $P$ per GPU, the communication complexity of both All-to-All and All-Gather within 1-bit-All-Reduce is $\frac{(n-1)P}{n}$, equivalent to the ideal 1-bit All-Reduce without overflow. In contrast, the complexity of \emph{All-Gather} in HiTopKComm is $(n-1)P$. This distinction demonstrates the significantly higher communication efficiency of Hierarchical-1-bit-All-Reduce ove HiTopKComm.
>
> In summation, the notion of hierarchy may appear intuitive and straightforward, with 1-bit-All-Reduce playing a pivotal role. This is evidenced by the recent influential work ZeRO++ [9], which adopts a scheme, called qgZ, nearly identical toHierarchical-1-bit-All-Reduce for large language model training.  ZeRO++ does not cite the paper on HiTopKComm, and its alignment with Hierarchical-1-bit-All-Reduce effectively varifies the efficacy and broader applicability of our approach.
>
>
> [1] Y. You, et al. "Large batch training of convolutional networks." arXiv:1708.03888, 2017.
>
> [2] S. Agarwal , et al. "On the utility of gradient compression in distributed training systems." MLSys, 2022.
>
> [3] T., Hanlin, et al. "1-bit Adam: Communication efficient large-scale training with Adam’s convergence speed." ICML, 2021.
>
> [4] S. Shi et al. "Towards scalable distributed training of deep learning on public cloud clusters." MLSys, 2021.
>
> [5] X. Hang, et al. "Compressed communication for distributed deep learning: Survey and quantitative evaluation." 2020.
>
> [6] X. Chen, et al. "Zo-adamm: Zeroth-order adaptive momentum method for black-box optimization." NeurIPS, 2019.
>
> [7] J. Zhuang, et al. "Adabelief optimizer: Adapting stepsizes by the belief in observed gradients." NeurIPS, 2020.
>
> [8] Y.Wang. et al."Communication-efficient adaptive federated learning." ICML, 2022.
>
> [9] G. Wang, et al. "ZeRO++: Extremely Efficient Collective Communication for Giant Model Training." arXiv:2306.10209, 2023.

---

> > ### Author Response · Authors · 2023-08-16
> > **Looking Forward to Further Discussion**
> >
> > Dear Reviewer,
> >
> > We highly appreciate your valuable comments. In our reponse,  we hope that your concerns have been addressed. We would like to discuss with you in this period and are willing to provide more according to your feedback or further questions.
> >
> > If you feel happy with our response, please consider updating your score. Feel free to reach us out in case you need any clarification.
> >
> > Thanks,
> >
> > Paper 5560 Authors

---

> > ### Comment · Reviewer_5r9C · 2023-08-18
> >
> > Thanks for the response. As the experimental results lack standard baselines, it is hard to justify the effectiveness of the proposed method in the current version. So I keep my original score.

---

> > > ### Author Response · Authors · 2023-08-19
> > > **Reponses to the Post-Rebuttal Feedback**
> > >
> > > We sincerely appreciate your valuable feedback. Our additional responses are outlined below:
> > >
> > > - We are eager to better comprehend your observation that "*the experimental results lack standard baselines*."  Could you kindly provide further clarification? As per our understanding, we have presented a comprehensive comparison between the proposed BRAM and strong baselines, uncompressed SGD/Adam. Notably, we have emphasized in both the paper and our prior responses that recent works demonstrated communication-compression optimizers remain less communication-efficient compared to uncompressed SGD/Adam in practical DNN training scenarios using DDP. *Given that we have selected the strongest contenders SGD/Adam as baselines in our experiments, we are puzzled by the notion that our experimental results lack standard baselines*.
> > >
> > > - Furthermore, we guess that you might still find the test accuracy of SGD when training ResNet-50 on ImageNet with a large batch size, dropping to 74.+%, somewhat unusual. Allow us to provide further clarification. **We conducted our experiments using the official Torchvision source code for training ResNet-50 (available at https://github.com/pytorch/vision/tree/main/references/classification) without any additional enhancements. The experimental settings adhered to common practices.**  It's important to note that the decline in test accuracy when training ResNet with a large batch size is a well-documented phenomenon, as evidenced in the LARS paper [1].  Crucially, *the reported 74.+% test accuracy for training ResNet-50 on ImageNet with an 8k batch size aligns perfectly with the results presented in a influential paper endorsed by the renowned scholar Kaiming He [2] (please refer to Table 1)*. In fact, if we were to apply the subtle warming technique outlined in [2] and the layer-wise scaling technique from LARS [1], we believe that BRAM with a large batch size could also achieve improved inference performance. However, we would like to reiterate that *our paper's primary objective is to achieve communication efficiency for practical distributed DNN training, rather than pursuing state-of-the-art inference performance in large batch size scenario*.
> > >
> > > - Addtionally,   we have extensively employed BRAM in training a diverse array of neural networks, spanning different architectures including convolutional-based, LSTM-based, and Transformer-based models. These applications encompass tasks in CV an NLP, ranging from normal training to retraining and fine-tuning. Notably, the recent advancement known as ZeRO++ [3] has adopted a method similar to BRAM, employing it for training large-scale LLMs. **These practical examples sufficiently demonstrat the broad effectiveness of BRAM**.
> > >
> > > [1] Y. You, et al. "Large batch training of convolutional networks." arXiv:1708.03888, 2017.
> > >
> > > [2] P. Goyal et al. "Accurate, large minibatch sgd: Training imagenet in 1 hour."  arXiv:1706.02677, 2017.
> > >
> > > [3] G. Wang, et al. "ZeRO++: Extremely Efficient Collective Communication for Giant Model Training." arXiv:2306.10209, 2023.

---

> > > > ### Author Response · Authors · 2023-08-21
> > > > **Eagerly Expecting a Last Minute Discussion**
> > > >
> > > > As the author-reviewer discussion deadline inchingly near, we are looking forward to engaging in a quick discussion with you. Please don't hesitate to reach us if you require any clarifications or further assistance.
> > > >
> > > > If our responses have addressed your concerns, we would greatly appreciate it if you could consider updating your rating.
> > > >
> > > >
> > > > Thanks,
> > > >
> > > > Paper #5560 Authors

---

### Official Review · Reviewer_gDkh · 2023-07-13

**Soundness:** 3 good
**Presentation:** 2 fair
**Contribution:** 3 good
**Rating:** 7
**Confidence:** 3

**Summary:**

The paper introduces a 1-bit adaptive optimizer called BRAM for distributed training of deep neural network models. BRAM combines the merits of SignSGD and Adam optimizers to achieve fast convergence speed, high inference performance, and reduced communication overhead. The proposed optimizer employs extreme 1-bit quantization for communication data while being lightweight in terms of computation. A hierarchical communication scheme called Hierarchical-1-bit-All-Reduce is also introduced to enhance communication efficiency. Extensive experiments on benchmark models, including ResNet-50 and BERT-Base, validate the effectiveness and efficiency of BRAM compared to uncompressed methods like SGDM/Adam and the 1-bit Adam optimizer. The experimental results demonstrate that BRAM achieves similar convergence rates to the baseline methods while significantly outperforming them in terms of training speed and system throughput, especially with a larger number of GPUs. Inference performance results show that BRAM performs comparably or even better than the baseline methods. Overall, BRAM offers a promising solution for distributed training with improved communication efficiency and convergence speed.

**Strengths:**

- The paper introduces BRAM, a 1-bit adaptive optimizer that combines the strengths of SignSGD and Adam. This optimizer effectively reduces communication overhead while maintaining fast convergence speed and high inference performance.

- The paper provides a theoretical convergence analysis for BRAM under certain assumptions. The analysis demonstrates that BRAM achieves a convergence rate comparable to uncompressed Adam, providing a solid theoretical foundation for its effectiveness.

- It proposes a hierarchical communication scheme called Hierarchical-1-bit-All-Reduce specifically designed for BRAM. This scheme leverages high intra-node bandwidth and reduces inter-node communication overhead, resulting in better scalability efficiency and improved system throughput.

- Extensive experimental validation: The paper conducts extensive experiments on benchmark models, including ResNet-50 and BERT-Base, LSTM, ViT etc., using different numbers of GPUs. The experimental results demonstrate the superiority of BRAM over uncompressed methods like SGDM/Adam and the 1-bit Adam optimizer in terms of training speed and system throughput, while maintaining comparable or even better inference performance.

**Weaknesses:**

- The theoretical convergence analysis of BRAM relies on several assumptions, such as bounded infimum, Lipschitz continuous gradient, and unbiased and independent noisy gradients. The sensitivity of BRAM's performance to these assumptions should be discussed to provide a more realistic assessment of its convergence properties.

- The paper briefly mentions the selection of hyperparameters such as the learning rate and exponential moving average factor, but there is limited discussion on the sensitivity of BRAM to these hyperparameters. A more thorough analysis and discussion of hyperparameter tuning would provide insights into achieving optimal performance. The paper does not include ablation studies to analyze the individual contributions of different components of BRAM. Investigating the impact of each component and analyzing their relative importance would enhance the understanding of the optimizer and provide insights for further improvements.

- Limited analysis of robustness to noisy gradients: The paper does not extensively discuss the robustness of BRAM to noisy gradients. While the theoretical analysis assumes unbiased and independent noisy gradients, a more in-depth investigation of BRAM's performance in the presence of varying levels of gradient noise would provide a comprehensive understanding of its behavior in practical scenarios.

- Although the paper mentions the reduced compression/decompression time of BRAM compared to 1-bit Adam, a more detailed analysis and comparison of the time required for compression and decompression would provide a better understanding of the overhead introduced by the quantization process.

- The paper focuses on communication efficiency but does not thoroughly discuss the memory efficiency of BRAM. Analyzing the memory requirements of BRAM compared to other optimizers would be valuable, especially in scenarios with limited memory resources.

**Questions:**

Please see the weaknesses.

**Limitations:**

There is no limitation discussion.

---

> ### Author Rebuttal · Authors · 2023-08-09
>
> We express our sincere gratitude for your volunteering time and your valuable comments. In response to your feedback, we have organized our response into the following points:
>
> **Q1.**  *The theoretical convergence analysis of BRAM relies on several assumptions, such as bounded infimum, Lipschitz continuous gradient, and unbiased and independent noisy gradients. The sensitivity of BRAM's performance to these assumptions should be discussed to provide a more realistic assessment of its convergence properties.*
>
> **R1.**  We greatly appreciate your efforts in highlighting this significant matter.  These assumptions  also serve as foundations for analyzing the theoretical convergence of prior adaptive optimizers, and they are common met during DNN training. Moreover,  they also serve as foundations for analyzing the theoretical convergence of prior adaptive optimizers. The theoretical impact of these assumptions on convergence is detailed in Equation (6) of Theorem 1. However, when considering their practical implications during real-world training, we acknowledge the inherent challenge of assessment, a aspect that also has not been addressed in prior research.
>
> **Q2.** *The paper briefly mentions the selection of hyperparameters such as the learning rate and exponential moving average factor, but there is limited discussion on the sensitivity of BRAM to these hyperparameters. A more thorough analysis and discussion of hyperparameter tuning would provide insights into achieving optimal performance. The paper does not include ablation studies to analyze the individual contributions of different components of BRAM. Investigating the impact of each component and analyzing their relative importance would enhance the understanding of the optimizer and provide insights for further improvements.*
>
> **R2.** Thank you for your valuable suggestion. Given the extensive number of experiments that needed to be carried out, we faced limitations in terms of the availability of numerous valuable GPUs (Nvidia A100-80G), over extended periods. Consequently, we opted not to employ grid search for hyperparameter selection. Instead, the hyperparameter values in BRAM followed the recommended value for Adam in official source codes or the paper on 1-bit Adam. Noting that the only exception is the value of $\beta$ in BRAM, fixed at $0.95$. Regarding the ablation experiments, we have conducted a comparison between BRAM and its uncompressed counterpart (SoftSignSGD). We appreciate your suggestion, and we intend to incorporate additional experiments to assess the sensitivity of these hyperparameters and gauge finer individual contributions, if numerous GPU resources become available in the future.
>
> **Q3.** *Limited analysis of robustness to noisy gradients: The paper does not extensively discuss the robustness of BRAM to noisy gradients. While the theoretical analysis assumes unbiased and independent noisy gradients, a more in-depth investigation of BRAM's performance in the presence of varying levels of gradient noise would provide a comprehensive understanding of its behavior in practical scenarios*
>
> **R3.** We appreciate your guidance on this matter.we have employed BRAM to train a diverse array of neural networks, including convolutional-based, LSTM-based, and Transformer-based models on various datasets with varying batch sizes. This methodology has provided an avenue for indirectly assessing the impact of varying levels of gradient noise. Thank your for this reminder, and we will add the direct robustness analysis to noisy gradient in the revised version.
>
> **Q4.** *Although the paper mentions the reduced compression/decompression time of BRAM compared to 1-bit Adam, a more detailed analysis and comparison of the time required for compression and decompression would provide a better understanding of the overhead introduced by the quantization process.*
>
> **R4.**  In fact, as depicted in Figure 4, we have included a comparison of the time required for compression/decompression of SGDM/Adam, 1-bit Adam, and BRAM,  and a comprehensive analysis of this aspect can be found in detail within Subsection 5.3.
>
> **Q5.** *The paper focuses on communication efficiency but does not thoroughly discuss the memory efficiency of BRAM. Analyzing the memory requirements of BRAM compared to other optimizers would be valuable, especially in scenarios with limited memory resources.*
>
> **R5.** The primary objective of this paper is to address the communication bottleneck issue in distributed training. As such, we have delved into a comprehensive analysis of communication efficiency within the main text. It's worth noting that we also touch upon memory efficiency in Section E of the Appendix. Owing to the presence of the error feedback state, BRAM exhibits less memory efficiency than Adam. However, as elaborated upon in Section E of the Appendix, the similarity in orders of magnitude between $m_t$ and $b_t$, coupled with the bounded range of $e_t$ within [-1, 1], enables BRAM to harness lower precision, such as FP16 or even FP8, to store the states $m_t$, $b_t$, and $e_t$, diverging from the FP32 state used in Adam. This, in turn, allows for the utilization of low-precision gradients to estimate updates. Consequently, BRAM presents a more memory-efficient potential than Adam. We are committed to further exploring and advancing this area in our future endeavors.

---

> > ### Author Response · Authors · 2023-08-16
> > **Looking Forward to Further Disccusion**
> >
> > Dear Reviewer,
> >
> > We highly appreciate your valuable comments. In our reponse,  we hope that your concerns have been addressed. We would like to discuss with you in this period and are willing to provide more according to your feedback or further questions.
> >
> > If you feel happy with our response, please consider updating your score. Feel free to reach us out in case you need any clarification.
> >
> > Thanks,
> >
> > Paper 5560 Authors

---

> > ### Author Response · Authors · 2023-08-17
> > **Concerns Regarding Sensitivity Experiments**
> >
> >
> >
> > We want to extend ourgratitude again for your valuable advice concerning the sensitivity experiments. As we explained in **R2**, we encountered limitations  in terms of the availability of numerous valuable GPUs over extended periods  during the paper preparation phase. Thus, conducting an exhaustive grid-search sensitivity analysis became unfeasible.   At present, we  still have access to only one machine with 8 GPUs (Nvidia V100); unfortunately, this constrained our ability to complete the sensitivity experiments  within the stipulated rebuttal period. Thanks to the extension of the author-reiviewer discussion period and our persistent experiment implementation, we now have the oppotunity to present the results  of the sensitivity experiments.
> >
> > **Table 1. Test Accuracy （ %）of BRAM with diffrent learning rates and a fixed exponential moving average factor (0.95) for training  ViT-B-16 on ImageNet**
> >  | $lr$ &emsp; |  &emsp;&emsp;   0.01  &emsp; &emsp;  | &emsp; &emsp;  0.02 &emsp;  &emsp;  | &emsp; &emsp; 0.03  &emsp; &emsp;  | &emsp; &emsp;  0.04  &emsp;&emsp;  | &emsp; &emsp;  0.05  &emsp;&emsp;  |
> > | -----------     | ----------- |-----------  | -----------  | -----------  |   -----------  |
> > |Test Acc &emsp;      |  &emsp; &ensp;   79.28  &emsp; &ensp;   |    &emsp; &emsp;   79.81 &emsp; &emsp;    |    &emsp; &emsp;  79.89 &emsp; &emsp;      |  &emsp; &emsp;    79.64  &emsp;  &emsp;    |    &emsp; &emsp;    78.85  &emsp;  &emsp;    |
> >
> >
> > **Table 2. Test Accuracy （ %）of BRAM with diffrent exponential moving average factors  and  fixed learning rate (0.03) for training  ViT-B-16 on ImageNet**
> >  | $\beta$ &emsp; |  &emsp;&emsp;   0.90  &emsp; &emsp;  | &emsp; &emsp;  0.92 &emsp;  &emsp;  | &emsp; &emsp;  0.95  &emsp; &emsp;  | &emsp; &emsp;  0.98  &emsp;&emsp;  | &emsp; &emsp;  0.99  &emsp;&emsp;  |
> > | -----------     | ----------- |-----------  | -----------  | -----------  | -----------  |
> > |Test Acc &emsp;      |  &emsp; &ensp;   79.88  &emsp; &ensp;   |    &emsp; &emsp;   79.92 &emsp; &emsp;    |    &emsp; &emsp;  79.89 &emsp; &emsp;      |  &emsp; &emsp;    79.95  &emsp;  &emsp;    | &emsp; &emsp;    79.81  &emsp;  &emsp;
> >
> > To evaluate the sensitivity of BRAM in training ViT-B-16 on ImageNet , we held $\beta$ constant at 0.95 and varied the values of the learning rate ($lr$), as well as maintained $\beta$ at 0.95 while altering the value of $\beta$ itself. As illustrated in Table 1-2,  BRAM exhibits reboustness to variations in the moving average factor, while it displays a slight sensitivity to adjustments in the learning rate. These sensitivity experiments will be thoughtfully incorporated into the revised version of the paper.
> >
> > Notably,  as we practiced in the paper,  simlpy aligning the learning rate of BRAM with the recommended values of Adam/1-bit Adam, while keeping $\beta$ fixed at 0.95, yields performance that is comparable to Adam in extensive experiments.
> >
> > We hope this adding information will be helpful to addresse some of your concerns.  Your thoughtful consideration of updating socre is greatly appreciated.

---

> > > ### Comment · Reviewer_gDkh · 2023-08-19
> > > **Post-rebuttal comment**
> > >
> > > Many thanks to the authors for their replies and additional results. The rebuttal has addressed most of my concerns, and I thus increase the rating and suggest acceptance.

---

> > > > ### Author Response · Authors · 2023-08-21
> > > > **Thank You for Rasing Rating**
> > > >
> > > > We are truly delighted that most of your concerns has been successfully addressed, and we want to express our sincere appreciation for the raising rating and the suggestion of acceptance for this paper.

---

### Official Review · Reviewer_sXik · 2023-07-26

**Soundness:** 3 good
**Presentation:** 3 good
**Contribution:** 2 fair
**Rating:** 5
**Confidence:** 2

**Summary:**

The paper proposes BRAM, a new 1-bit optimizer that directly quantizes the entire adaptive update. It theoretically proves that BRAM promises the same convergence speed as the full-precision Adam even with extreme 1-bit quantization. Considering the difference between intra-node and inter-node bandwidth, the paper also proposes Hierarchical-1-bit-All-Reduce,  a hierarchical communication scheme that aggregates our 1-bit data by leveraging the ultra-high intra-node bandwidth. The experiments show that BRAM can achieve empirical high performance on top of Hierarchical-1-bit-All-Reduce.

**Strengths:**

1. The presentation is good. The writing is clear and easy to follow.
2. The paper makes a detailed evaluation and analysis of 1 Adam.
3. The appendix proves that the element-wise quantization function is unbiased.

**Weaknesses:**

1. It is unclear whether using the same hyperparameters for both the exponential moving averages of gradients and the squared gradients would have negative effects.
2. BRAM has less room for further optimization and can only be employed in very few applications because it sends updates directly back.

**Questions:**

Can BRAM be applied to pipeline parallel distributed training of foundation models?

**Limitations:**

The authors do not mention the limitations of their work.

---

> ### Author Rebuttal · Authors · 2023-08-09
>
> We express our sincere gratitude for your volunteering time and your valuable comments. In response to your feedback, we have organized our response into the following points:
>
> **Q1.**  *It is unclear whether using the same hyperparameters for both the exponential moving averages of gradients and the squared gradients would have negative effects.*
>
> **R1.** Thank you for bringing this matter to our attention.  Drawing from both our theoretical analysis and empirical findings,  it becomes evident that sharing the same moving average factor $\beta$ for $m_t$ and $b_t$  for BRAM can still guarantee the same theoretical convergence rate as Adam with different moving average factors, and the experiments also demonstrate it achieves s a similar training speed and comparable, or even superior, inference performance when compared to Adam. In summary, sharing the moving average factor $\beta$ for $m_t$ and $b_t$  for BRAM does not lead to negative effects. Instead, it bestows  tangible advantages,  eliminating the necessity for bias correction and significantly reducing the amount of required tuning work.
>
> **Q2.** *BRAM has less room for further optimization and can only be employed in very few applications because it sends updates directly back.*
>
> **R2.** Thank you for your valuable feedback. It appears you might have inferred a constraint on the application of BRAM due to the difficult implementation of the entire update rather than the gradient. However, I would like to clarify that this implementation is actually straightforward through a communication hook in DDP of PyTorch. Further details, including code examples, can be found in the uploaded zip file. As highlighted in our paper, we have extensively applied BRAM to train a wide range of neural networks, encompassing various architectures such as convolutional-based, LSTM-based, and Transformer-based models. These applications cover diverse computer vision (CV) and natural language processing (NLP) tasks, including normal training, retraining, and fine-tuning. Through these practical examples, we have demonstrated BRAM's user-friendly nature and its versatility.
>
> **Q3.** *Can BRAM be applied to pipeline parallel distributed training of foundation models?*
>
> **R3.** The latest influential work ZeRO++ [1] (it posted on arXive subsequent to our NeuRIPS submission ) incorporated a similar method (gqz) to Hierarchical-1-bit-All-Reduce of BRAM,  employing it to reduce the communication overhead across inter-nodes during the training of giant foundational models. This implementation exemplifies the expansive potential of BRAM for foundational model training. As we emphasized in the paper,  BRAM is applicable to data parallel for distributed training, which is orthogonal to pipeline parallel.
>
> [1] G. Wang, et al. "ZeRO++: Extremely Efficient Collective Communication for Giant Model Training." arXiv:2306.10209, 2023.

---

> > ### Author Response · Authors · 2023-08-16
> > **Looking Forward to Further Discussion**
> >
> > Dear Reviewer,
> >
> > We highly appreciate your valuable comments. In our reponse,  we hope that your concerns have been addressed. We would like to discuss with you in this period and are willing to provide more according to your feedback or further questions.
> >
> > If you feel happy with our response, please consider updating your score. Feel free to reach us out in case you need any clarification.
> >
> > Thanks,
> >
> > Paper 5560 Authors

---

> > > ### Comment · Reviewer_sXik · 2023-08-19
> > >
> > > Thanks to the authors and other reviewers for their replies and discussions. I read them carefully and it helped me understand the work better. I'll keep the rating at 5.

---

### Author Rebuttal · Authors · 2023-08-09


I greatly appreciate the efforts put forth by the chairs and reviewers to ensure high-quality reviews. We have studied the comments and suggestions seriously and carefully, which have definitely helped us improve the work. We now respond to the reviewers' comments as follows. The reviewer comments are presented below in italicized font, and our responses are provided in normal font.

Before addressing the reviewers' comments, we would like to reiterate the novelties of our work:

-  We propose a novel 1-bit  optimizer, dubbed  BRAM. To the best of our knowledge, it is the first native communication-compression adaptive optimizer that element-wise quantizes the entire model update, making compression/decompression computationally light and the extreme quantization ratio exert its best function.

- We theoretically prove that despite emolying extreme 1-bit quantization is employed,  BRAM still promise the same convergence speed as the full-precision Adam.

- We develop a new communication scheme for 1-bit communication, called Hierarchical-1-bit-All-Reduce, which sufficiently harnesses the ultra-fast intra-connects to accelerate the local communication, and utilize more efficient commutation primitives to further reduce the communication overhead.

- We perform extensive distributed  training experiments to demonstrate the effectiveness of the proposed algorithm. As far as we know, running with DDP,  our algorithm is the first work to consistently trump SGDM/Adam  in terms of entire running time at little/no inference performance cost}, reaching up to $2.47\times$ speedup for ResNet-50 and ${6.26}\times$ speedup for BERT-Base on $64$ GPUs.

It is important to note that our work primarily focuses on enhancing communication efficiency without compromising test accuracy compared to the baseline SGDM/Adam. Our emphasis is not on pursuing state-of-the-art performance, but rather on achieving improved efficiency in distributed training.

---

### Decision · Program_Chairs · 2023-09-21

**Decision:**

Accept (poster)

**Comment:**

The paper proposes a 1-bit adaptive optimizer for distributed learning. In discussion, most of the reviewers tended towards accepting the paper, and I agree with that consensus. The main issue was the lack of comparison to multiple compressed ADAM baselines, but I don't view this as being a bar to acceptance.